# *SF3B1* mutation–mediated sensitization to H3B-8800 splicing inhibitor in chronic lymphocytic leukemia

Irene López-Oreja[1,2,3,4], André Gohr[2], Heribert Playa-Albinyana[1,4], Ariadna Giró[1], Fabian Arenas[1,4], Morihiro Higashi[1], Rupal Tripathi[1], Mònica López-Guerra[1,3,4], Manuel Irimia[2,6,7], Marta Aymerich[1,3,4], Juan Valcárcel[2,6,7], Sophie Bonnal[2], Dolors Colomer[1,3,4,5]

Splicing factor 3B subunit 1 (SF3B1) is involved in pre-mRNA branch site recognition and is the target of antitumor-splicing inhibitors. Mutations in *SF3B1* are observed in 15% of patients with chronic lymphocytic leukemia (CLL) and are associated with poor prognosis, but their pathogenic mechanisms remain poorly understood. Using deep RNA-sequencing data from 298 CLL tumor samples and isogenic *SF3B1* WT and K700E-mutated CLL cell lines, we characterize targets and pre-mRNA sequence features associated with the selection of cryptic 3′ splice sites upon *SF3B1* mutation, including an event in the *MAP3K7* gene relevant for activation of NF-κB signaling. Using the H3B-8800 splicing modulator, we show, for the first time in CLL, cytotoxic effects in vitro in primary CLL samples and in *SF3B1*-mutated isogenic CLL cell lines, accompanied by major splicing changes and delayed leukemic infiltration in a CLL xenotransplant mouse model. H3B-8800 displayed preferential lethality towards *SF3B1*-mutated cells and synergism with the BCL2 inhibitor venetoclax, supporting the potential use of SF3B1 inhibitors as a novel therapeutic strategy in CLL.

## Introduction

Splicing factor 3B subunit 1 (SF3B1) is a core component of the spliceosome, the cellular machinery in charge of removing introns from precursor mRNAs (pre-mRNA splicing). The spliceosome recognizes specific sequences at the intron boundaries, namely, the 5′ splice site (5′ss) and the 3′ splice site (3′ss), the latter comprising a branch point (BP) adenosine located 18–35 nucleotides (nt) upstream of the conserved AG dinucleotide at 3′ end of the intron and a polypyrimidine tract (Py tract) located between the BP and AG. Alternative usage of splice sites, namely, alternative splicing (AS), importantly contributes to protein diversity (Wahl et al, 2009; Gallego-Paez et al, 2017). SF3B1 recognizes the branch site adenosine (Cretu et al, 2016) and *SF3B1* mutations identified in cancer cells promote the usage of cryptic 3′ss located 10–30 nucleotides upstream of the canonical 3′ss. Each 3′ss has been reported to use its own BP adenosine (Darman et al, 2015; DeBoever et al, 2015; Alsafadi et al, 2016). *SF3B1* mutations have been detected in various cancer types including myelodysplastic syndromes (in 28–65% of the cases) (Malcovati et al, 2011; Papaemmanuil et al, 2011), chronic lymphocytic leukemia (CLL) (in 7–25% of the cases) (Quesada et al, 2011; Wang et al, 2011), uveal melanoma (in 15–22% of the cases) (Furney et al, 2013; Harbour et al, 2013; Martin et al, 2013), and other solid tumors (Seiler et al, 2018a) and have been proposed to contribute to different aspects of tumor development and progression (Dolatshad et al, 2016; Lee et al, 2018; Dalton et al, 2019; Liu et al, 2021; Lieu et al, 2022). In CLL, the frequency of *SF3B1* mutations increases with disease evolution and treatment (Rossi et al, 2011; Baliakas et al, 2015; Leeksma et al, 2019), and a hotspot mutation region has been identified between the fifth and eighth Huntington elongation factor 3 PR65/A TOR (HEAT) domain repeats of the protein, encoded by exons 14–16, K700E being the most frequent mutation (Wan & Wu, 2013). *SF3B1* mutations are often subclonal, and their prevalence is associated with poor prognosis (Landau et al, 2013; Nadeu et al, 2016). Patients harboring the mutation present a shorter time to first treatment (TTFT), a shorter progression-free survival, and overall survival (OS) (Quesada et al, 2011; Wang et al, 2011; Jeromin et al, 2014; Stilgenbauer et al, 2014; Nadeu et al, 2016; Mansouri et al, 2023). Even patients treated with newly approved therapies, such as ibrutinib and venetoclax, seem to have a worse outcome (Byrd et al, 2019; Roberts et al, 2019), emphasizing the need for novel therapeutic approaches.

Interestingly, several antitumor drugs targeting SF3B1 have been identified and developed in the last decade (Bonnal et al, 2012; Webb et al, 2013; León et al, 2017). Among them, H3B-8800 is an orally bioavailable small molecule that binds to the SF3B complex

[1]Institut d'Investigacions Biomèdiques August Pi i Sunyer, Barcelona, Spain   [2]Centre for Genomic Regulation, The Barcelona Institute of Science and Technology, Barcelona, Spain   [3]Hematopathology Section, Department of Pathology, Hospital Clínic, Barcelona, Spain   [4]Centro de Investigación Biomédica en Red en Oncologia, Madrid, Spain   [5]Universitat Barcelona, Barcelona, Spain   [6]Universitat Pompeu Fabra, Barcelona, Spain   [7]Institució Catalana de Recerca i Estudis Avançats, Barcelona, Spain

Correspondence: dcolomer@clinic.cat; sophie.bonnal@crg.eu; juan.valcarcel@crg.eu

and modulates splicing. This compound displays preclinical anti-tumor activity in myeloid malignancies with preferential lethality towards cancer cells harboring splicing factor mutations (Seiler et al, 2018b), with an acceptable safety profile (NCT02841540) (Steensma et al, 2021).

Here, we characterize the changes in alternative splicing induced by *SF3B1* mutations in CLL patients and in an isogenic CLL MEC1 cell line bearing the most frequent *SF3B1* cancer mutation (K700E). We also characterize the cytotoxic effect of H3B-8800 both in vitro and in vivo and we explore a novel therapeutic approach combining H3B-8800 with venetoclax, a commonly used drug in CLL treatment.

# Results

To study the impact of *SF3B1* mutations in CLL, we developed the first isogenic MEC1 cell lines harboring (or not) the most prevalent *SF3B1* mutation in CLL, K700E. MEC1 cells were edited using CRISPR/Cas9 and single-stranded oligodeoxynucleotides (ssODN) harboring two stabilizing phosphorothioate linkages at their 5′ and 3′ ends (Renaud et al, 2016). We replaced c.2098A>G to induce a lysine to glutamic acid amino acid change at position 700. In addition, we introduced a silent mutation replacing XGG PAM sequence by XGA (c.2106G>A), to prevent its further recognition after initial editing (Paquet et al, 2016) (Fig 1A). A WT editing control harboring the PAM mutation but retaining lysine at position 700 was also generated. From a total of 68 clones, we obtained three clones bearing heterozygous K700E and PAM site mutations, detected by PCR and cleavage with Taqα1 restriction enzyme (Fig 1B), validated by Sanger sequencing (Fig 1C) and digital PCR (Fig 1D), confirming about 50% variant allelic frequency (VAF) in *SF3B1*$^{K700E}$ clones. Western blot analyses showed that the levels of SF3B1 protein were not affected by the presence of the K700E mutation (Fig 1E). Of relevance, we observed that *SF3B1*$^{K700E}$ clones displayed significantly lower viability and metabolic activity than *SF3B1*$^{WT}$ clones (Fig 1F and G).

## Alternative splicing changes associated with *SF3B1*$^{K700E}$ mutation

To investigate the impact of *SF3B1* mutation on splicing in CLL, we first used RNA-seq data from tumor B cells from 298 patients diagnosed with CLL from the International Cancer Genome Consortium (ICGC) (Puente et al, 2015) and B cell samples from four healthy donors. 190 patients were from our institution (Hospital Clínic Barcelona) and 108 were provided by other institutions of the Spanish ICGC consortium. Biological and clinical information of these patients are provided in Table S1. The toolset *vast-tools* (Tapial et al, 2017) and the "percent spliced in" (PSI) metric were used to quantify alternative exons (ES), intron retention (IR), alternative 3′ (Alt3′ss), and alternative 5′ (Alt5′ss) splice sites usage in each patient. Principal component analysis (PCA) on all quantified events (minimum coverage of 10 reads) showed an heterogeneous profile related to the sampling process, as reported previously (Dvinge et al, 2014) (Fig 2A). PCA of Alt3′ss events showed that principal component 2 (variance 7%) was driven by the proportion of tumor cells carrying *SF3B1* mutation cancer cell fraction (CCF) (Fig 2B). Although samples with very low CCF (CCF < 12%) mainly

clustered with *SF3B1*$^{WT}$ samples, samples with a high CCF (CCF > 50%) formed a different cluster, with the exception of two samples harboring *SF3B1* mutations outside of the domain involved in branch point recognition (E862K and M757T, Fig 2C), which clustered with *SF3B1*$^{WT}$ samples (Fig 2B). In addition, samples with low CCF (CCF > 12 and <50) formed an intermediate cluster (Fig 2B). PCA of Alt5′ss, IR, and SE events did not show any specific clustering related to *SF3B1* mutational status (Fig 2D–F). For further analyses, we selected 225 *SF3B1* WT cases and 15 cases harboring *SF3B1* mutations (8 K700E, 3 H662D, 2 R625C, 1 L743F, and 1 T663I) with high CCF (≥50%), excluding samples with mutations E862K and M757T in *SF3B1*, in other splicing factors or in other genes related to RNA metabolism (see the Materials and Methods section). PCA and hierarchical clustering in *SF3B1*-mutated samples did not reveal global transcriptomic changes associated with specific mutations, although the comparison between 8 *SF3B1* K700E samples and 3 *SF3B1* H662D mutations did identify 136 differential alternative splicing events (53 IR, 42 Alt3′ss, 33 SE, and 8 Alt5′ss), and 162 differentially expressed genes, suggesting specific molecular effects of individual mutations (Fig S1A–E, Tables S2, and S3).

To further study the effects of *SF3B1* mutations, additional RNA-seq analyses included RNAs from MEC1 *SF3B1*$^{K700E}$ heterozygous and WT isogenic cell lines described above. The fraction of differentially spliced events (see the Materials and Methods section) between *SF3B1* WT and *SF3B1*-mutant samples was enriched in Alt3′ss events both in CLL samples or MEC1 isogenic cell lines compared with the distribution of detected AS events (Fig 3A, Tables S4, and S5). Although all the differentially spliced AS events identified in CLL samples were observed in the cell lines, only 211/776 differentially spliced AS events identified in cell lines were detected in CLL cases (Fig 3A). Moreover, the highest correlation between ΔPSI values from CLL samples and cell lines was observed for Alt3′ss (R = 0.9), followed by IR (R = 0.7) and SE (R = 0.58) and tailed by Alt5′ss (R = 0.31) (Fig 3B). 41% of these differentially spliced AS events in patients or cell lines were predicted to disrupt the open reading frame (ORF) (Fig 3C). Consistent with previous work linking ORF disruption and NMD-mediated mRNA degradation (Leeksma et al, 2021), these events were associated with a significant decrease in expression of the corresponding genes (Mann–Whitney test, P < 0.0001) (Fig 3D). Both in patient samples and cell lines, AS events changing upon *SF3B1* mutation were enriched in genes involved in biological processes such as mRNA catabolism and splicing, chromatin organization, histone modification, cell cycle, glucose metabolism, apoptotic signaling, and cytoplasmic microtubule organization, and with B cell-relevant pathways including NF-κB, TGFßR, and MAPKK signaling pathways, phosphatidylinositol biosynthesis, and immunological synapse formation (Fig 3E).

## *SF3B1*$^{K700E}$-induced alternative and cryptic 3′ splice sites display specific sequence features

To identify novel (not annotated) and cryptic (only used in *SF3B1*-mutated cases) splicing events, we developed *JuncExplorer*, a software designed to detect and quantify the usage of annotated and non-annotated splice junctions (see the Materials and Methods section). Using this tool, events with a single pair of 3′ss choices (compared with events involving 3 or more 3′ss choices) were the

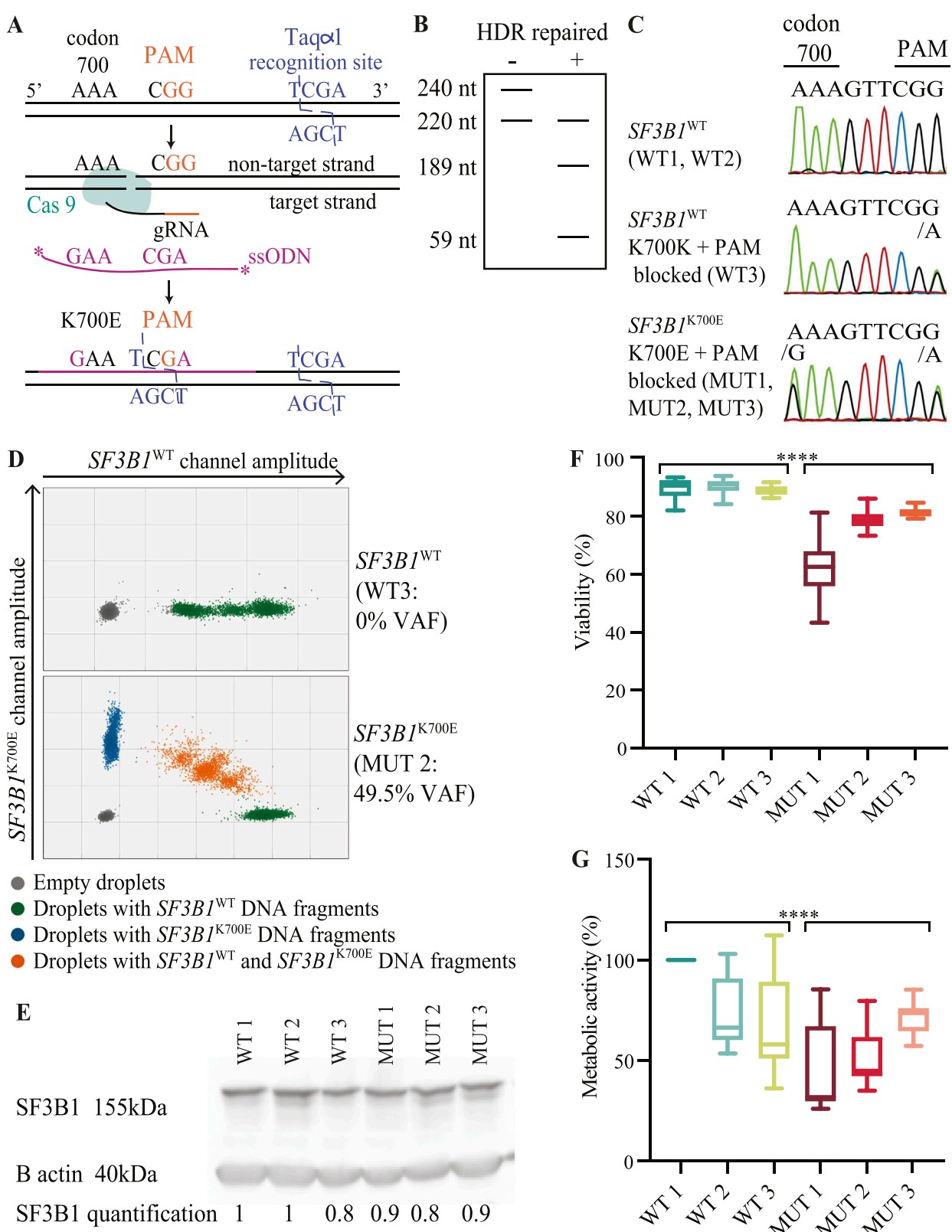

**Figure 1. Establishment and characterization of *SF3B1*[WT] and *SF3B1*[K700E] MEC1 chronic lymphocytic leukemia isogenic cell lines.**
**(A)** Representation of the CRISPR–Cas9 splicing factor 3B subunit 1 (*SF3B1*) K700E knock-in strategy, indicating the position of codon 700, PAM sequence, and Taqα1 recognition site, location of sgRNA, and donor single-stranded oligodeoxynucleotides with nucleotide changes to induce the K700E mutation and PAM sequence editing. **(B)** Expected size of PCR products corresponding to homology-directed repair clones after digestion with Taqα1 restriction enzyme, depending on absence (−) or presence

most frequently mis-regulated type of Alt3'ss between $SF3B1^{WT}$ and mutated samples, both in patients and in cell lines, (Fig S2, Tables S6, and S7). 3'ss choices were classified as "canonical" when they corresponded to the most used 3'ss in $SF3B1^{WT}$ samples, whereas other 3'ss where classified as alternative or cryptic Alt3'ss, depending on whether their use (PSI) in $SF3B1^{WT}$ samples was >5 (alternative Alt3'ss) or ≤5 (cryptic Alt3'ss), respectively. A change in 3'ss usage |ΔPSI| ≥15 upon $SF3B1$ mutation was required for classification as either alternative or cryptic Alt3'ss. We defined a total of 595 alternative Alt3'ss events and 268 cryptic Alt3'ss events in cell lines and 38 alternative Alt3'ss and 15 cryptic Alt3'ss in patients' samples. Although most non-canonical AGs were located upstream and near of the canonical 3'ss (Fig 4A, Tables S8, and S9) as previously described (Darman et al, 2015), we also detected cryptic and alternative 3'ss at much longer distances and downstream of the canonical 3'ss. Therefore, we categorized four groups depending on the location of the 3'ss with increased usage upon $SF3B1$ mutations (UpF: Upstream Far; UpN: Upstream Near; DoN: Downstream Near; DoF: Downstream Far) (Fig 4B). Activation of adjacent 3'ss either upstream or downstream from the canonical 3'ss (usually referred to as NAGNAG) (Hiller & Platzer, 2008) represented two additional (rather minor) categories. Examples of each group were independently validated by RT–PCR in RNA samples from MEC1 $SF3B1^{WT}$ and $SF3B1^{K700E}$ cells (Fig S3A and B, Table S10). We used Matt software (Gohr & Irimia, 2019) to analyze sequence features associated with cryptic, alternative, and canonical splice sites (Fig 4C). Alternative and, especially, cryptic 3'ss are generally weaker 3'ss than their canonical counterparts, regardless of their relative location, and display shorter predicted BP to AG distances (Figs 4C and S4A). These results suggest that $SF3B1$ mutations facilitate the recognition of weaker 3'ss sequences. Moreover, canonical 3'ss associated with distant Alt3'ss tend to be stronger than those associated with closer Alt3'ss, suggesting that activation of distant sites requires efficient initial interaction of their canonical sites with the splicing machinery (Fig S4B). In summary, our results expand the catalogue and relative location/distances of cryptic and alternative 3'ss activated by $SF3B1$ mutations.

An important, still poorly explored, mechanistic question is whether non-canonical 3'ss activation upon $SF3B1$ mutation requires the use of alternative branch point sequences. To investigate this question, we carried out mutagenesis analyses using expression plasmids (minigenes) in which the 3'ss region of an Adenovirus Major Late intron was replaced by 3'ss regions of $ZNF561$ (displaying an Alt3'ss 13 nucleotides downstream of the canonical site; DoN event), $MAP3K7$ (displaying an Alt3'ss 20 nucleotides upstream of the canonical site; UpN event) or $TARBP1$ (displaying a NAGNAG event) (Fig 4D and Table S11). WT plasmids or plasmids harboring mutations in predicted BP sequences for each construct

were transfected in $SF3B1^{WT}$ or $SF3B1^{K700E}$ MEC1 cells and the use of canonical and non-canonical 3'ss was measured (Fig 4E–G). The three sets of constructs recapitulated activation of the Alt3'ss in $SF3B1^{K700E}$ cells. In the case of $ZNF561$, the use of each 3'ss appears to be associated with a different BP because mutation of the 5'-most BP (BP2 mutant) led to activation of the non-canonical 3'ss in $SF3B1^{WT}$ cells, whereas mutation of the 3'-most BP (BP1 mutant) prevented activation of the non-canonical 3'ss in $SF3B1^{K700E}$ cells (Fig 4E). In the case of $MAP3K7$, in contrast, the 3'-most BP appeared to be sufficient to mediate the selection of the canonical 3'ss in $SF3B1^{WT}$ cells and the switch to the non-canonical 3'ss in $SF3B1^{K700E}$ cells. Specifically, its mutation (BP1 mutant) reduced the use of the non-canonical 3'ss in both cell types, whereas mutation of the 5'-most BP (BP2 mutant) did not significantly alter the choice of 3'ss in either of the two cell lines. Additional mutation of other putative branch sites were included leading to a further decrease in the usage of the alternative AG in both cell types, without affecting canonical AG usage (Figs 4F and S5A and B) or splicing efficiency (Fig S5C) consistent with previous report (Li et al, 2021). In the case of the increased relative use of an alternative downstream, NAGNAG site in $TARBP1$, the 3'-most BP appears to be relevant for activation of the Alt3'ss because its mutation (BP1 mutant) prevented its increased relative use in $SF3B1^{K700E}$ cells, whereas the 5'-most BP does not seem to have an impact on NAGNAG use (Figs 4G and S5D). We conclude that the interplay between two different BP sequences is important for Alt3'ss use in $ZNF561$ whereas it is not for the $MAP3K7$ or $TARBP1$ introns.

## Identification of cryptic splice sites as surrogate markers for $SF3B1$ mutations in CLL

Following up on these findings and to find aberrant isoforms that could be used as surrogate markers for the functional impact of $SF3B1$ mutations, we selected the events with ΔPSI ≥50 in $SF3B1$-mutated patient samples as quantified by $JuncExplorer$. One 3'ss in the $ZDHHC16$ gene (junction ID: chr10:97454799-97455639(+)) with a ΔPSI = 50.93 showed also the lowest average PSI (2.87) in $SF3B1^{WT}$ patient samples (Wilcoxon test, $P = 6.71 \times 10^{-10}$). This Alt3'ss event was validated by RT-qPCR (Fig S6). Although detection of the canonical isoform was possible in all tested samples (39 $SF3B1$ WT samples, 29 $SF3B1$-mutated samples with high CCF [CCF ≥ 50], 11 with low CCF [50 > CCF ≥ 12], and 9 with very low CCF [CCF < 12]), the cryptic $ZDHHC16$ isoform was only detected in mutated samples with CCF >12 (except for cases when $SF3B1$ was mutated in E862K or M757T) and three samples with CCF < 12 (Fig S6A and B). These results confirmed that cryptic $ZDHHC16$ isoform detection can be used as a surrogate marker to identify $SF3B1$ mutations that alter the profile of 3'ss selection in CLL patient samples.

---

(+) of edited sites. **(C)** DNA sequence of CRISPR/Cas9-modified MEC1 clones. The nucleotides corresponding to SF3B1 codon 700 and Cas9-targeted PAM sequence are shown. WT1 refers to parental MEC1 cell line, WT2 to a WT clone which underwent clonal selection after failed CRISPR/Cas9 edition, WT3 to a $SF3B1^{K700K}$ clone with the silent mutation in PAM, and MUT1-3 to the three $SF3B1^{K700E}$ clones. **(D)** Quantification by digital PCR of MEC1 $SF3B1^{WT}$ and $SF3B1^{K700E}$ clones. Variant allele frequency is the fraction abundance between droplets matching the specific K700E variant divided by the overall number of droplets analyzed. **(E)** SF3B1 protein expression in $SF3B1^{WT}$ and $SF3B1^{K700E}$ MEC1 cell lines. The quantification was calculated by normalizing the SF3B1/β actin ratio to sample WT1. **(F)** Viability of $SF3B1^{WT}$ and $SF3B1^{K700E}$ MEC1 cell lines (n = 17 replicates for each group) after 48 h in culture at 30,000 cell/100 µl concentration, assessed by flow cytometry (Annexin V-, PI-). **(G)** Metabolic activity of $SF3B1^{WT}$ and $SF3B1^{K700E}$ MEC1 cell lines (n = 12 replicates for each group) after 48 h in culture at 30,000 cell/100 µl concentration assessed by MTT assay. **(F, G)** Data information: in (F, G) data are represented as mean ± SD. Mann–Whitney test (****$P$ < 0.0001). Source data are available for this figure.

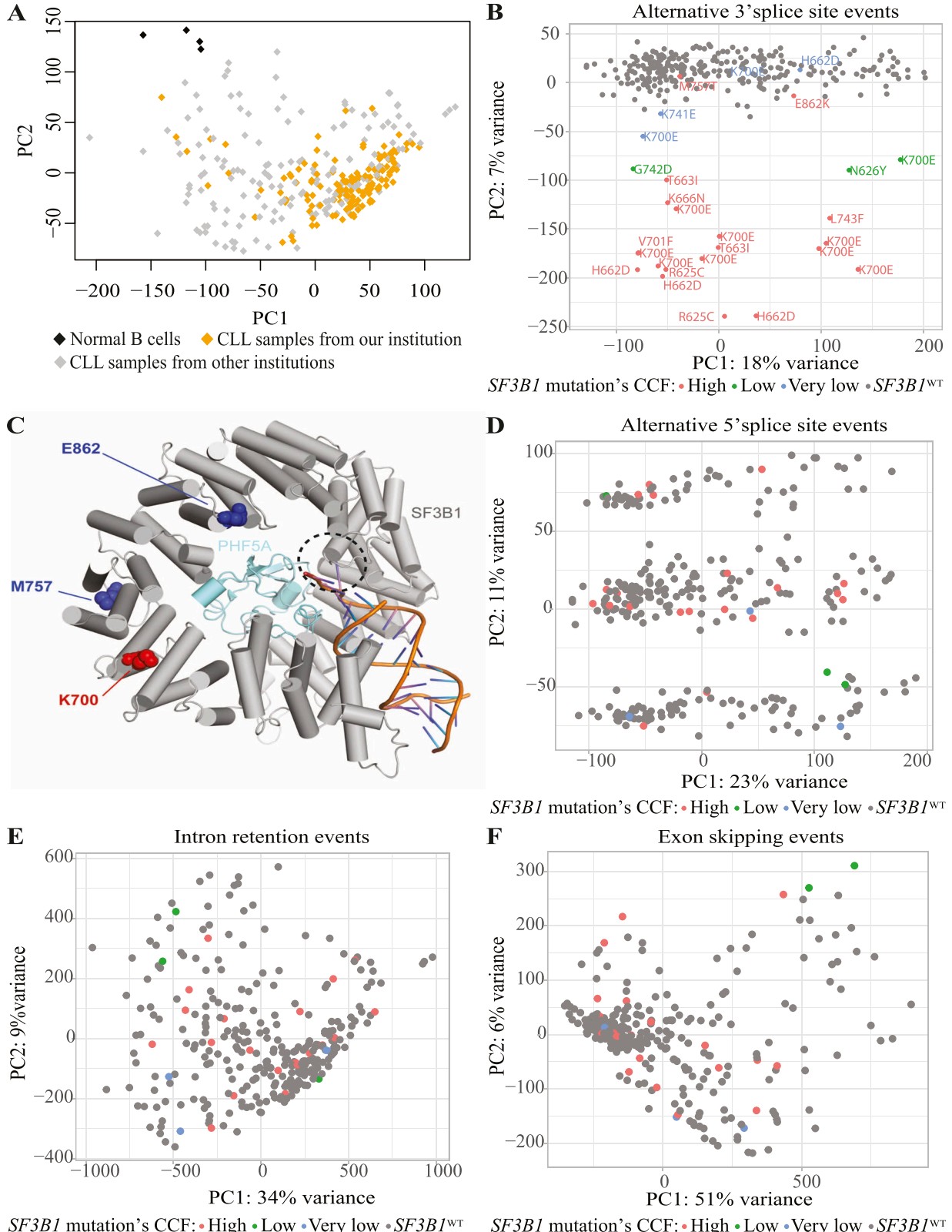

**Figure 2. Principal component analysis (PCA) of AS events found in IGCG chronic lymphocytic leukemia (CLL) RNA-seq data.**
**(A, B)** PCA calculated for the AS events (>10 reads coverage supporting the percent spliced in calculation) found in PBMCs from 298 CLL cases and four healthy donors (herein referred to as normal B cells). Samples from our institution were labeled in yellow, and in grey the samples sent to us for processing from other institutions of the Spanish International Cancer Genome Consortium. **(B, D, E, F)** PCA calculated for Alt3'ss (B), Alt5'ss events (D), IR events (E), and ES events (F) found in CLL cases. PCA was

### SF3B1 mutation affects AS events relevant for activation of the NF-κB–signaling pathway

At the functional level, five genes (*LRRK2*, *MAP3K12*, *MAP3K7*, *MAP3K10*, and *TGFBR1*) involved in the activation of MAPKK were differentially spliced in *SF3B1*-mutated patients' samples (Table S12). Recently, MAP3K7 AS has been postulated to promote NF-κB–driven tumorigenesis (Lee et al, 2018; Liu et al, 2021). Interestingly, we identified an alternative 3'ss (HsaALTA1027057-2/4 in VastDB) in the *MAP3K7* gene itself displaying strong mis-regulation upon *SF3F1* mutation (ΔPSI = 60 in *SF3B1*$^{WT}$ versus *SF3B1*$^{K700E}$ CLL patients' samples and ΔPSI = 70 in *SF3B1*$^{WT}$ versus *SF3B1*$^{K700E}$ MEC1 CLL cell lines). This 3'ss is located upstream of the canonical 3'ss of exon 5, which encodes part of the kinase domain, and its usage results in the disruption of the coding sequence and degradation of the corresponding transcript by NMD (Li et al, 2021). We validated this event by RT–qPCR in *SF3B1*$^{WT}$ versus *SF3B1*$^{K700E}$ MEC1 CLL isogenic cell lines and in 32 CLL patients' samples (Fig 5A). We also observed decreased MAP3K7 protein expression in *SF3B1*$^{K700E}$ MEC1 CLL cell lines compared with *SF3B1*$^{WT}$ cells (Fig 5B), in line with the AS event targeting the transcript to NMD degradation (Li et al, 2021). In addition, an alternative 3'ss (HsaALTA1015253-2/3, VastDB nomenclature) downstream of the canonical 3'ss of *NFKB1* exon 11, a gene of the NF-κB–signaling pathway, was also detected in the RNA-seq data from CLL patients' samples and validated in MEC1 cell lines (Fig S3A). Enrichment of differential gene expression and differentially spliced events between *SF3B1*$^{WT}$ and *SF3B1*$^{K700E}$ MEC1 CLL cell lines showed a significant enrichment of genes involved in I-κB kinase/NF-κB signaling and its regulation (GO:0043122, GO:0007249) (Fig 5C). Measurement of NF-κB activity in *SF3B1*$^{WT}$ and *SF3B1*$^{K700E}$ cell lines showed a trend towards an increase of the active p65 form and of the ratio between phosphorylated p65 and total p65 in *SF3B1*$^{K700E}$ samples, although they did not reach statistical significance (Fig 5D and E).

### H3B-8800 exerts cytotoxic effect in SF3B1-mutated CLL samples

We first assessed the cytotoxic effect of the splicing modulator H3B-8800 in the isogenic *SF3B1*$^{K700E}$ and *SF3B1*$^{WT}$ MEC1 CLL cell lines. Cytotoxicity was measured after incubation of these cell lines with 1–100 nM H3B-8800 for 48 h. H3B-8800 induced cytotoxicity in a dose-dependent manner in both *SF3B1*$^{WT}$ and *SF3B1*$^{K700E}$ cell lines, this effect being significantly enhanced in *SF3B1*$^{K700E}$ cells at doses higher than 25 nM (mean viability at 75 nM was 52.1% ± 12.3% for *SF3B1*$^{K700E}$ cells and of 71.8% ± 10.8% for *SF3B1*$^{WT}$ cells) (Fig 6A).

Primary CLL samples were also treated with H3B-8800 as above. Given the difficulty to detect changes in AS associated with *SF3B1* mutations in samples with low *SF3B1*-mutant CCF (Fig 2B), *SF3B1*-

mutated samples were divided into two groups according to their CCF > 80% and CCF < 80%. We found a high heterogeneous response among CLL samples, being the mean viability compared with untreated cells in mutated CLL cases with CCF > 80% of 70% ± 10.3%, lower than the viability observed in mutated CLL samples with CCF < 80% (79% ± 12.7%) and in *SF3B1*$^{wt}$ CLL cells (81% ± 11.9%) (Fig 6B and Table S13).

To examine the cytotoxic properties of H3B-8800 in the context of the microenvironment, we carried out co-culture experiments of primary CLL cells or MEC1 cell lines with either bone marrow–derived mesenchymal (HS-5) or human follicular dendritic cell-like (HK) cell lines. We observed that in the presence of HS-5 or HK cells, H3B-8800 induced lethality in both *SF3B1*$^{K700E}$ and *SF3B1*$^{WT}$ CLL primary cells (Fig 6C and D) and MEC1 cell lines (Fig 6E and F) with a prominent effect in *SF3B1*$^{K700E}$ cell lines (Fig 6F). These data suggest that H3B-8800 is active even in the presence of a protective microenvironment.

### H3B-8800 delays leukemic infiltration in vivo in NSG mouse model

To test the efficacy of H3B-8800 in vivo, we generated luciferase-expressing MEC1 *SF3B1*$^{K700E}$ and *SF3B1*$^{WT}$ cell lines. Cells were intravenously inoculated via the tail in NOD–SCID interleukin-2 receptor gamma (*IL2Rγ*)$^{null}$ (NSG) mice, forming xenografts of isogenic MEC1 cells with *SF3B1*$^{WT}$ or *SF3B1*$^{K700E}$ (n = 16 and n = 17, respectively). After 10 d, upon detection of bioluminescence, mice began to be treated orally with vehicle or H3B-8800 (6 mg/Kg) daily for 10 d and luciferase activity was captured at days 3, 6, and 10 by bioluminescence in vivo imaging. We observed lower bioluminescence signal or leukemic infiltration in *SF3B1*$^{K700E}$ MEC1 cell line compared with *SF3B1*$^{WT}$ MEC1 cell line and a decrease of this signal after H3B-8800 treatment in both of them (Fig 7A and B). After 10 d of treatment, leukemic infiltration in bone marrow and peripheral blood and spleen and liver weight were assessed upon euthanizing the mice. The mean values of bone marrow leukemic infiltration (measured by the presence of CD19$^+$ B lymphocytes by flow cytometry) were 54% ± 13.9% and 41% ± 17.4% for the *SF3B1*$^{WT}$ and *SF3B1*$^{K700E}$ groups treated with vehicle, respectively (Fig 7C). H3B-8800 treatment led to a reduction in CD19$^+$ bone marrow leukemic infiltration only in the *SF3B1*$^{K700E}$ group (87% of reduction, Fig 7D), indicating a preferential effect of the drug on *SF3B1*$^{K700E}$ cells. As the leukemic infiltration in peripheral blood was rather low (8% in *SF3B1*$^{WT}$ groups and 2.6% in *SF3B1*$^{K700E}$ groups, Fig 7E) a reduction of CD19 infiltration (from 8% to 3%) upon H3B-8800 treatment was only detected in the *SF3B1*$^{WT}$ group (Fig 7F). H3B-8800 treatment decreased the weight of liver and spleen in both groups (Fig 7G and H) and spleen size (Fig 7I). To detect B lymphocyte infiltration in the spleen, we performed

calculated on the percent spliced in values of Alt3'ss, Alt5'ss, IR, and ES events of each sample using *ggfortify* and *ggplot2* packages in R (v3.6.3). The mutational status and cancer cell fraction (CCF) (very low, CCF < 12%; low, CCF 12–50%; high, CCF > 50%) is indicated by color code as defined in each panel. **(C)** Location of splicing factor 3B subunit 1 (SF3B1) residues that induce (K700) or not (M757, E862) alterations in patterns of 3'ss usage upon mutation in the 3D structures of SF3B1 (in grey) and PHF5A (SF3B7) (in light blue). Highlighted in dark blue are residues that do not lead to altered 3'ss usage upon mutation (M575T, E862K) and in red K700, a mutation hotspot (frequently K700E) which induces characteristic patterns of 3'ss usage. mRNA and U2snRNA are represented in orange (base pairing corresponds to annealing between U2 snRNA BP recognition sequence and BP nucleotides flanking the BP adenosine, which bulges out from the helix) and the binding site of drugs such as SSA, pladienolide B, or H3B-8800 is marked by the circle with dotted line. The structural representation was prepared using the PyMOL Molecular Graphics System, Version 1.3r1 (Schrodinger, LLC) based on PDB 6FF4 structure from the Protein Data Bank.

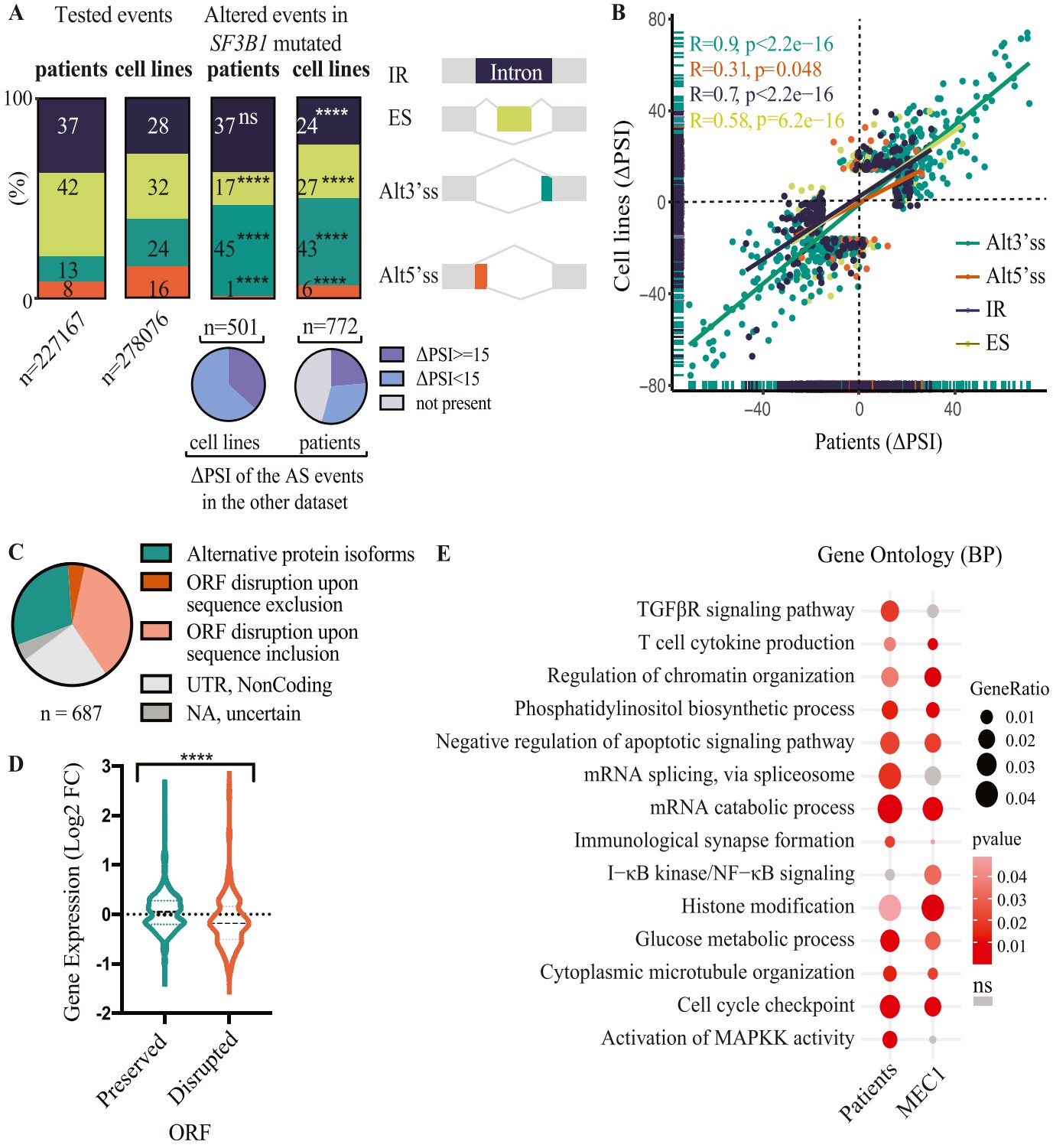

**Figure 3. Differentially alternatively spliced events found in chronic lymphocytic leukemia (CLL) samples and *SF3B1*<sup>WT</sup> and *SF3B1*<sup>K700E</sup> MEC1 CLL isogenic cell lines.**
**(A)** Number of differentially spliced (|ΔPSI| ≥ 15) AS event types associated with *SF3B1* mutations in primary CLL cases and *SF3B1*<sup>K700E</sup> MEC1 cell lines compared with all assessed AS events using *vast-tools*. The pie charts at the bottom represent the fraction of the AS events found in cell lines or CLL samples which are also observed in the other dataset. Test of equal proportion (****P-value < 0.0001; ns, not significant). **(B)** Correlation between AS events ΔPSI values comparing *SF3B1* mutant versus WT samples found in MEC1 isogenic cell lines and in CLL samples, with a |ΔPSI| ≥ 15 at least in one of the datasets. Pearson correlation coefficients were estimated for each AS type as indicated. **(C)** Effects on the mRNA of differentially spliced (|ΔPSI| ≥ 15) transcripts found in CLL or MEC1 cell lines using *vast-tools*. The pie charts represent the fraction of events corresponding to each of the categories of predicted impact on ORF, located in UTR or unknown. **(D)** Comparison of gene expression changes associated with the events predicted to preserve or disrupt ORFs in CLL patient samples or MEC1 cell lines. The statistical significance between both groups was evaluated using a Mann–Whitney test (****P-value < 0.0001). **(E)** *SF3B1* mutation induces AS changes in genes involved in diverse biological processes. Enriched Gene Ontology terms

CD79a immunohistochemical staining and found that massive tumor infiltration was observed in the *SF3B1*^WT group compared to *SF3B1*^K700E group. These results agree with the lower leukemic infiltration observed by luminescence analysis (Fig 7A and B) and reported in other studies using other cancer cell lines (Wang et al, 2016). A reduction of the infiltration assessed by CD79a staining was observed in both groups after H3B-8800 treatment (Fig 7J). These results indicate that *SF3B1*^K700E CLL MEC1 cells display lower infiltration capacity than *SF3B1*^WT cells and are particularly sensitive to H3B-8800 treatment, at least in terms of inhibition of bone marrow infiltration.

### H3B-8800 induces intron retention and exon skipping in H3B-8800–treated *SF3B1*^WT and *SF3B1*^K700E MEC1 cell lines

To analyze the differential impact of H3B-8800 on alternative splicing in *SF3B1*^K700E versus *SF3B1*^WT MEC1 CLL cells, we treated four million cells of each genotype in triplicate with 75 nM H3B-8800 or DMSO (control) for 6 h. RNA was isolated and deep sequenced using Illumina pair-ended sequencing technology and the datasets analyzed using *vast-tools* software. We observed that H3B-8800 induced extensive intron retention and exon skipping changes in both cell lines, with nearly 8,000 introns detected as more retained and more than 10,000 exons as more skipped upon treatment (Figs 8A and S7A and B), which were enriched in genes involved in cell signaling, apoptosis, and RNA splicing (Fig 8B). The results of H3B-8800 treatment on IR or ES were largely overlapping for *SF3B1*^WT and *SF3B1*^K700E cells (Fig S7A and B), suggesting that the mutational status of *SF3B1* does not seem to exert a major influence on the overall impact of H3B-8800 on these events. In contrast, a larger fraction of distinct effects of the drug on *SF3B1*^WT and *SF3B1*^K700E cells was observed when analyzing Alt5'ss and Alt3'ss events (Fig S7C and D), with nearly as many common events as *SF3B1*^WT- or *SF3B1*^K700E-specific events.

Next, we analyzed sequence features associated with AS changes induced by the drug in *SF3B1*^WT cells using *Matt* software (Gohr & Irimia, 2019). We first compared the group of regulated exons with different control groups (Fig 8C and see the Materials and Methods section). We observed that exons skipped upon H3B-8800 treatment are shorter than those in all control groups (Fig 8C). Interestingly, compared with alternative non-changing and cryptic exons, skipped exons display stronger 5'ss and 3'ss (maximum entropy score for human model) and the surrounding introns and exons are shorter. In contrast, compared with constitutive exons (PSI > 95 in both groups), skipped exons display weaker 5'ss and 3'ss and the surrounding introns are longer (Fig 8C). These results argue that H3B-8800–sensitive exons display characteristic genomic features distinct from those typical of regulated alternative exons.

Second, we compared retained introns upon H3B-8800 treatment (n = 8,171) with introns that were spliced more efficiently (n = 62) and with introns that did not change (n = 675) upon drug treatment. We observed that (i) retained introns and their flanking exons are shorter than the ones from non-changing introns, (ii) their 5'ss and 3'ss are weaker (Fig 8D), and (iii) display higher GC content (GCC), whereas better spliced introns display lower GCC, at least in their 5' half (Fig 8E). A representation of GCC along model genomic regions showed that retained introns are located in a more general GC-rich genomic context, with similar GC content in introns and exons, suggesting the prevalence of intron definition over exon definition mechanisms during splice site recognition of these introns (Amit et al, 2012) (Fig 8E).

The special susceptibility of *SF3B1*-mutated cells to H3B-8800 might be explained by synergistic or antagonistic effects of drug treatment and *SF3B1* mutation on particular splicing alterations. Out of 252 events associated with *SF3B1* mutation, 56 were reversed by H3B-8800 treatment, whereas 68 were exacerbated ($|\Delta PSI| > 10$) (Fig 8F). For example, in the event identified in MAP3K7, Alt3'ss selection was slightly reduced ($\Delta PSI = -9.89$) after H3B-8800 treatment.

### H3B-8800 shows synergistic effects with venetoclax

RNA-seq analysis of H3B-8800–treated MEC1 cell lines identified 26 AS events in genes involved in apoptosis that were differentially spliced ($|\Delta PSI| \geq 15$) between *SF3B1*^K700E and H3B-8800–treated *SF3B1*^K700E cells (Fig S8). 23/26 of these spliced events also changed upon treatment of *SF3B1*^WT cells with the drug. One of these events was the well-described cassette exon 2 of the MCL1 gene (HsaEX0038237, VastDB nomenclature) (Bae et al, 2000) which switches between anti- and pro-apoptotic isoforms upon drug treatment (Fig 9A) and has been proposed to mediate the cytotoxic effects of SF3B1 inhibitors (Gao & Koide, 2013; Xargay-Torrent et al, 2015; Larrayoz et al, 2016). H3B-8800 treatment (75 nM) led to similar exon 2 skipping effects in *SF3B1*^WT ($\Delta PSI = -37.6$) and *SF3B1*^K700E ($\Delta PSI = -34$) cell lines (Fig 9B). Similar effects were observed in WT and K700E-mutant CLL primary cells (Fig 9C). As BCL2 inhibitors have been shown to display synergistic effects with splicing modulators (Hacken et al, 2018; Aird et al, 2019), we incubated *SF3B1*^WT or *SF3B1*^K700E MEC1 cell lines with venetoclax (0–250 nM), a selective inhibitor of the anti-apoptotic protein BCL2, and H3B-8800 (0–75 nM) for 48 h. The results showed synergistic effects between the two drugs in the two cell lines (Fig 9D and E). These data were used to calculate a zero interaction potency synergy score using *SynergyFinder* 2.0 software (Ianevski et al, 2020) (Fig 9F). We obtained zero interaction potency synergy scores around 30 for *SF3B1*^WT and *SF3B1*^K700E cells, suggesting synergistic effects (scores >10 are considered to reveal likely synergistic interactions between two drugs). Altogether, these results demonstrate that the combination of H3B-8800 with venetoclax results in enhanced cytotoxic effect.

---

(*enrichGO* R package) for biological processes in genes displaying differentially spliced events in *SF3B1*-mutated CLL patient samples and *SF3B1*^K700E MEC1 cell lines. GO terms are indicated along with the statistical significance (*P*-value, red circles, ns, not significant) and gene ratio that indicates the ratio of genes with an AS event from all the tested genes (black circles). Data information: IR, intron retention; ES, exon skipping; Alt3'ss and Alt5'ss, alternative 3'ss and alternative 5'ss.

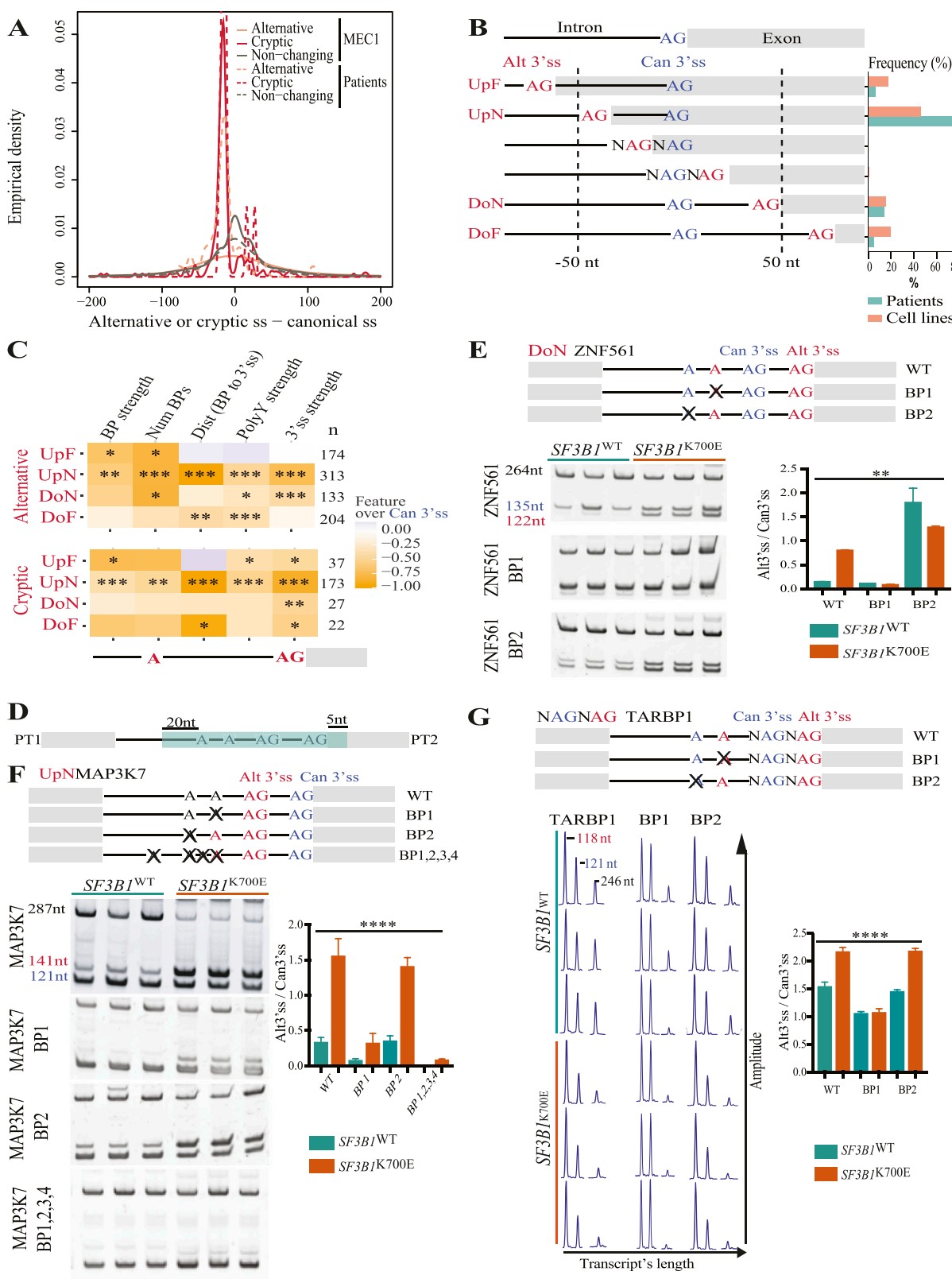

**Figure 4. Location, features, and mechanisms of regulation of cryptic and Alt3'ss activated by splicing factor 3B subunit 1 (*SF3B1*) mutations.**
**(A)** Empirical density of distances (in nucleotides) between cryptic or Alt3'ss and their canonical 3'ss found in *SF3B1*-mutated MEC1 cell lines (continuous lines) and *SF3B1*-mutated chronic lymphocytic leukemia patient samples (dashed lines). A negative distance indicates that the cryptic 3'ss or Alt3'ss is located upstream of the canonical 3'ss counterparts. Distribution of distances is shown for cryptic and for Alt3'ss with |ΔPSI| > 15% and range of 5%, and non-changing Alt3'ss with |ΔPSI| < 1%, as

# Discussion

Genome-wide studies have highlighted the genomic complexity and interpatient heterogeneity of CLL (Landau et al, 2015; Puente et al, 2015; Knisbacher et al, 2022), arguing for a high number of low-frequency driver genes contributing to disease progression. *SF3B1* is one of the most recurrently mutated genes, present in about 15% of CLL patients. These mutations—typically subclonal and subject to clonal evolution—have been linked to specific biological features of the disease such as unmutated imumoglobulin (Ig) heavy-chain variable region (IGHV) CLL subgroup, the intermediate CLL epigenetic subgroup, stereotyped BCR subset 2, R110-mutated IGLV3-21 subgroup, or 11q deletion (Baliakas et al, 2015; Queirós et al, 2015; Nadeu et al, 2021) and have been associated with more aggressive disease and poorer clinical outcome (Rossi et al, 2013; Jeromin et al, 2014; Stilgenbauer et al, 2014; Baliakas et al, 2015; Nadeu et al, 2017; Takahashi et al, 2018).

SF3B1 is an essential core component of spliceosomal U2 snRNP complexes and is directly involved in pre-mRNA branch point recognition. Previous work identified changes in alternative splicing associated with *SF3B1* mutation (Darman et al, 2015; Alsafadi et al, 2016; Kesarwani et al, 2017; Gupta et al, 2018). Our PCA of RNA-seq data from the ICGC CLL project (Puente et al, 2015) showed specific clustering of mutations and alternative 3'ss usage only in patients carrying clonal mutations located in the C-terminal HEAT repeats domain of the protein, which directly contacts branch point sequences, but not in samples harboring mutations located outside of the domain (E862K and M757T). *SF3B1* clonal mutations with VAF > 12% are predictive of shorter TTFT, whereas samples with subclonal mutations mainly clustered with *SF3B1*$^{WT}$ samples and are not predictive of shorter TTFT (Nadeu et al, 2016). In contrast, in the COMPLEMENT I clinical trial where chlorambucil or chlorambucil and ofatumumab treatments were compared, patients harboring tumors with low VAF clones showed shorter progression-free survival (Tausch et al, 2020). These results highlight the need to report and consider VAF values when analyzing the results of clinical trials in the era of new targeted therapies.

To cleanly dissect the molecular and cellular effects of *SF3B1*$^{K700E}$, we developed isogenic MEC1 CLL cell lines carrying or not the most frequent (K700E) *SF3B1* mutation using CRISPR/Cas9-genome editing. Although several *SF3B1*-mutant isogenic cell lines were previously generated from B-cell precursor leukemia cells, myeloid leukemia cells, murine embryonic stem cells, or induced pluripotent stem cells (Darman et al, 2015; Murthy et al, 2019; Dalton et al, 2019), our cell lines are the first isogenic CLL cell lines harboring WT and mutated *SF3B1* reported so far. Our results confirmed that these cell lines mimic the effect of *SF3B1* mutations on splicing regulation observed in patient samples and can therefore be used for further research on the molecular and cellular consequences of *SF3B1* mutations in CLL. As *SF3B1* mutations are associated with worse prognosis for CLL patients, the lower viability of *SF3B1* K700E MEC1 cell lines compared with WT isogenic cells appears counterintuitive. It should be kept in mind, however, that *SF3B1* mutations have been associated with cellular senescence in murine models and that CLL-like disease only develops in the context of associated *ATM* mutations (Yin et al, 2019). Therefore, it seems likely that such additional mutational events are required in combination with *SF3B1* mutations to produce an aggressive cancer phenotype.

Because the first reports of *SF3B1* mutations and aberrant splicing in cancer (Furney et al, 2013; Landau et al, 2013; Ferreira et al, 2014; Gentien et al, 2014), important efforts have been made to characterize their AS profiles, highlighting the activation of cryptic 3'ss typically located ≈15–24 nucleotides upstream of the canonical 3'ss (Darman et al, 2015; DeBoever et al, 2015; Alsafadi et al, 2016; Wang et al, 2016) and—less frequently—Alt'3ss located at longer distances (Kesarwani et al, 2017) or closely downstream of the canonical 3'ss (Darman et al, 2015; Alsafadi et al, 2016; Wang et al, 2016). Our analysis of CLL samples and isogenic CLL cell lines confirm and expand these observations and identify intronic features that can contribute to mediate the effects of *SF3B1* mutations. Using novel software, we classified Alt3'ss into four groups depending on the location (upstream/downstream) and the distance (≥/<50 nt) to the canonical 3'ss. Interestingly, we observed,

---

indicated. **(B)** Schematic representation of subtypes of cryptic and Alt3'ss patterns relative to the canonical 3'ss (upstream near, UpN: ≤ 50 nucleotides upstream; upstream far, UpF: > 50 nucleotides upstream; downstream near, DoN: ≤ 50 nucleotides downstream; downstream far, DoF: > 50 nucleotides downstream) and their relative frequencies in *SF3B1*-mutated chronic lymphocytic leukemia samples and *SF3B1*$^{K700E}$ MEC1 cell lines. **(C)** Comparison between the normalized means of a variety of intronic features calculated using Matt software (Gohr & Irimia, 2019) for each type of cryptic or Alt3'ss. The mean of the features of the predicted BP, cryptic, and Alt3'ss compared with those of canonical 3'ss and predicted associated BP: BP strength (BP score for the best predicted BP), Num BPs (number of predicted BPs), Dist BP to 3'ss (distance from the best predicted BP to the 3'ss), PolyY strength (polypyrimidine tract score for the best-predicted BP), 3'ss strength (maximum entropy score of 3'ss using a model trained with human splice sites) (Yeo & Burge, 2004; Gohr & Irimia, 2019). Orange rectangles in the heatmap indicate that cryptic/Alt3'ss display lower values compared with canonical 3'ss for each of the features compared. The schematic representation below the heatmap indicates the position of the features examined within the architecture of 3'ss regions. **(D)** Schematic representation of the adenovirus major late transcript where different Alt3'ss were introduced to replace its natural 3'ss region. The insert in green includes the regions with alternative 3'ss AGs, from 20 nucleotides upstream of the 5' branch point adenosine (A) to five nucleotides downstream of the 3' AG. PT1 and PT2 correspond to sequences used for detection of the non-spliced and spliced products by RT–PCR. **(E, F, G)** Schematic representation of WT and mutant Alt3'ss regions corresponding to ZNF561 (E), MAP3K7 (F), or TARBP1 (G) included in the minigene constructs with the predicted branch site adenosines and 3'ss. Single-point mutations in BP1 or in BP2 are indicated with an "X." **(E, F)** Lower left panel: results of RT–PCR analysis of RNAs from *SF3B1*$^{WT}$ or *SF3B1*$^{K700E}$ MEC1 cell lines transfected with the indicated ZNF561 (E) or MAP3K7 (F) minigenes, analyzed by polyacrylamide gel electrophoresis. Lower-right panel: quantification of the ratio between the Alt3'ss and canonical 3'ss usage of the amplification products shown in the left panel. The product corresponding to intron retention migrates at 264/287 nucleotides, whereas products corresponding to the usage of alternative 3'ss are annotated in blue (canonical 3'ss) and red (alternative 3'ss) with their respective sizes in the left panels. **(G)** Lower-left panel: fragment analysis of RT–PCR products from RNAs of *SF3B1*$^{WT}$ or *SF3B1*$^{K700E}$ MEC1 cell lines transfected with the indicated TARBP1 minigenes. Lower-right panel: ratio between the Alt3'ss and canonical 3'ss percent spliced in quantification area under the curve of the peaks (e.g., Alt3'ss percent spliced in: aberrant peak/[aberrant peak + canonical peak]) shown in the left panel. The product corresponding to intron retention migrates at 246 nucleotides, whereas products corresponding to the usage of alternative 3'ss are annotated in blue (canonical 3'ss) and red (alternative 3'ss). Data information: green bars: *SF3B1*$^{WT}$ cells. Orange bars: *SF3B1*$^{K700E}$ cells. ZNF561 minigene replicates: WT = 3, BP1 = 3, BP2 = 3. MAP3K7 minigene replicates: WT = 9, BP1 = 6, BP2 = 6, BP1, 2, 3, 4 = 3. TARBP1 minigene replicates: WT = 9, BP1 = 6, BP2 = 6. Kruskal–Wallis test (**$P$ < 0.01, ****$P$ < 0.0001).
Source data are available for this figure.

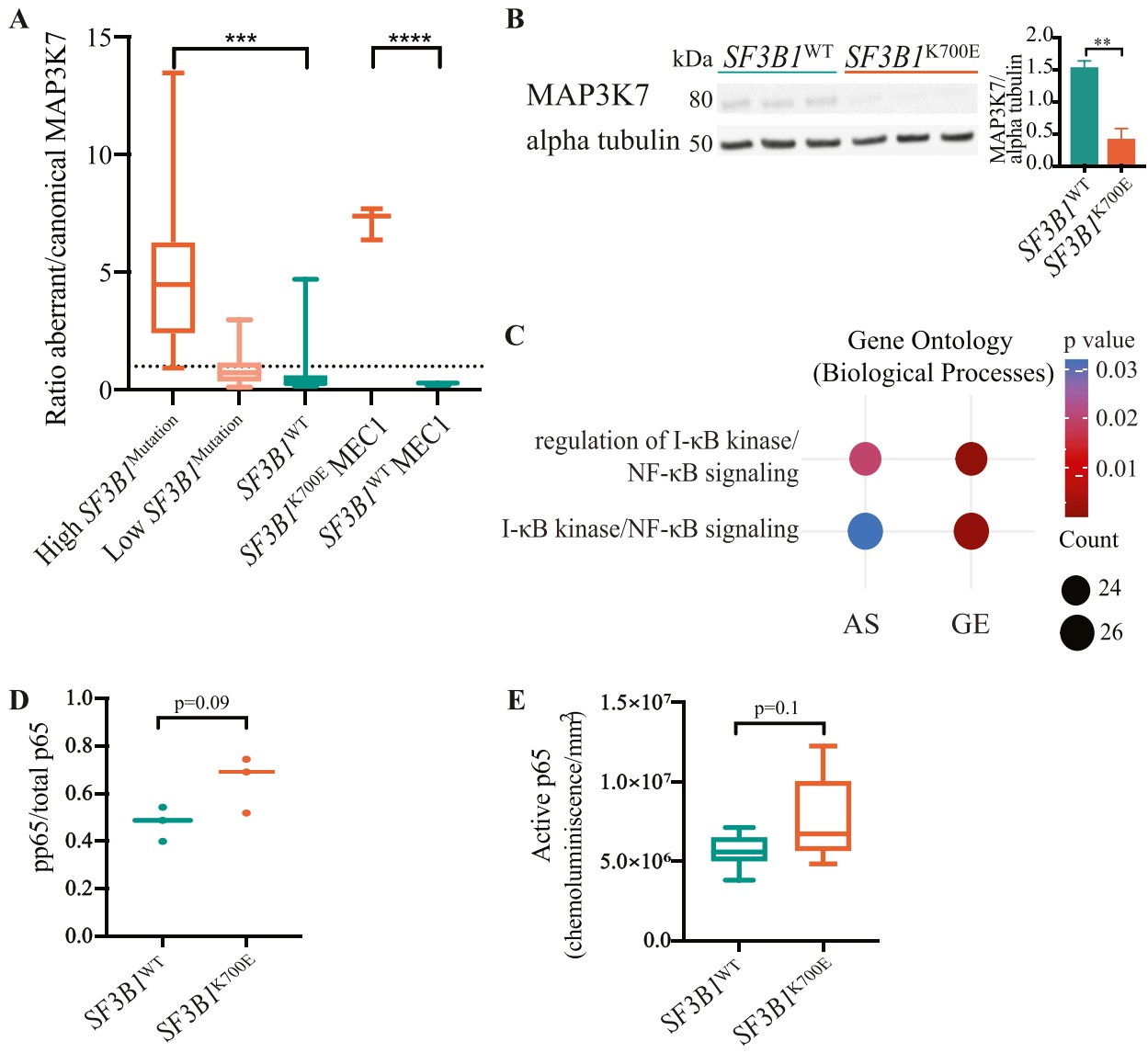

**Figure 5. Impact of SF3B1[K700E] on AS events of genes involved in the NF-κB pathway.**
**(A)** Ratio between Alt3'ss and canonical MAP3K7 isoforms measured by RT–qPCR in patients' samples and in *SF3B1*[WT] versus *SF3B1*[K700E] MEC1 chronic lymphocytic leukemia cell lines. High *SF3B1* mutation corresponds to cancer cell fraction > 50%. Low *SF3B1* mutation corresponds to cancer cell fraction < 50%. **(B)** Western blot analysis of MAP3K7 expression, using alpha tubulin as a loading control protein. Quantification was carried out using Multi Gauge software and normalized by calculating the MAP3K7/alpha tubulin ratio. **(C)** Enriched Gene Ontology terms (*enrichGO* R package) for genes displaying differentially spliced events or differences in gene expression in *SF3B1*[K700E] MEC1 cell lines. GO terms are indicated along with the statistical significance (*P*-value, red and blue circles) and gene counts (black circles). **(D)** Western blot quantification of phosphorylated p65 (pp65) levels relative to total p65 amounts in nuclear extracts of *SF3B1*[WT] and *SF3B1*[K700E] MEC1 chronic lymphocytic leukemia cell lines. **(E)** p65 DNA binding of the same samples as in (D) measured using an ELISA-based chemiluminescence kit. Data information: statistical significance was estimated using an unpaired *t* test (\*\**P*-value < 0.01; \*\*\**P*-value < 0.001; \*\*\*\**P*-value < 0.0001).

both in CLL primary cells and CLL cell lines that cryptic 3'ss can be located not only upstream but also downstream of the canonical 3'ss and that the distance between the two 3'ss, both in upstream and downstream cases, can be longer than 50 nucleotides and therefore is not as constrained as previously thought.

We observed that alternative and cryptic 3'ss typically display weaker BP, Py tract, and 3'ss than their associated canonical 3'ss, consistent with previous reports (Darman et al, 2015; Alsafadi et al, 2016). These properties are common regardless of alternative/cryptic 3'ss location, although cryptic 3'ss features are typically

weaker than alternative 3'ss features, suggesting that very weak 3'ss cannot be recognized by WT SF3B1 but are recognized by SF3B1[K700E]. The effects of mutating predicted branch points in minigene constructs were consistent with the use of different branch points by the alternative 3'ss and the canonical 3'ss, as previously described (Darman et al, 2015; Alsafadi et al, 2016; Tang et al, 2016; Carrocci et al, 2017). However, our results suggest more complex scenarios as well. For example, in contrast with the activation of upstream branch points and cryptic 3'ss reported in uveal melanoma (Alsafadi et al, 2016), the activation of a downstream cryptic

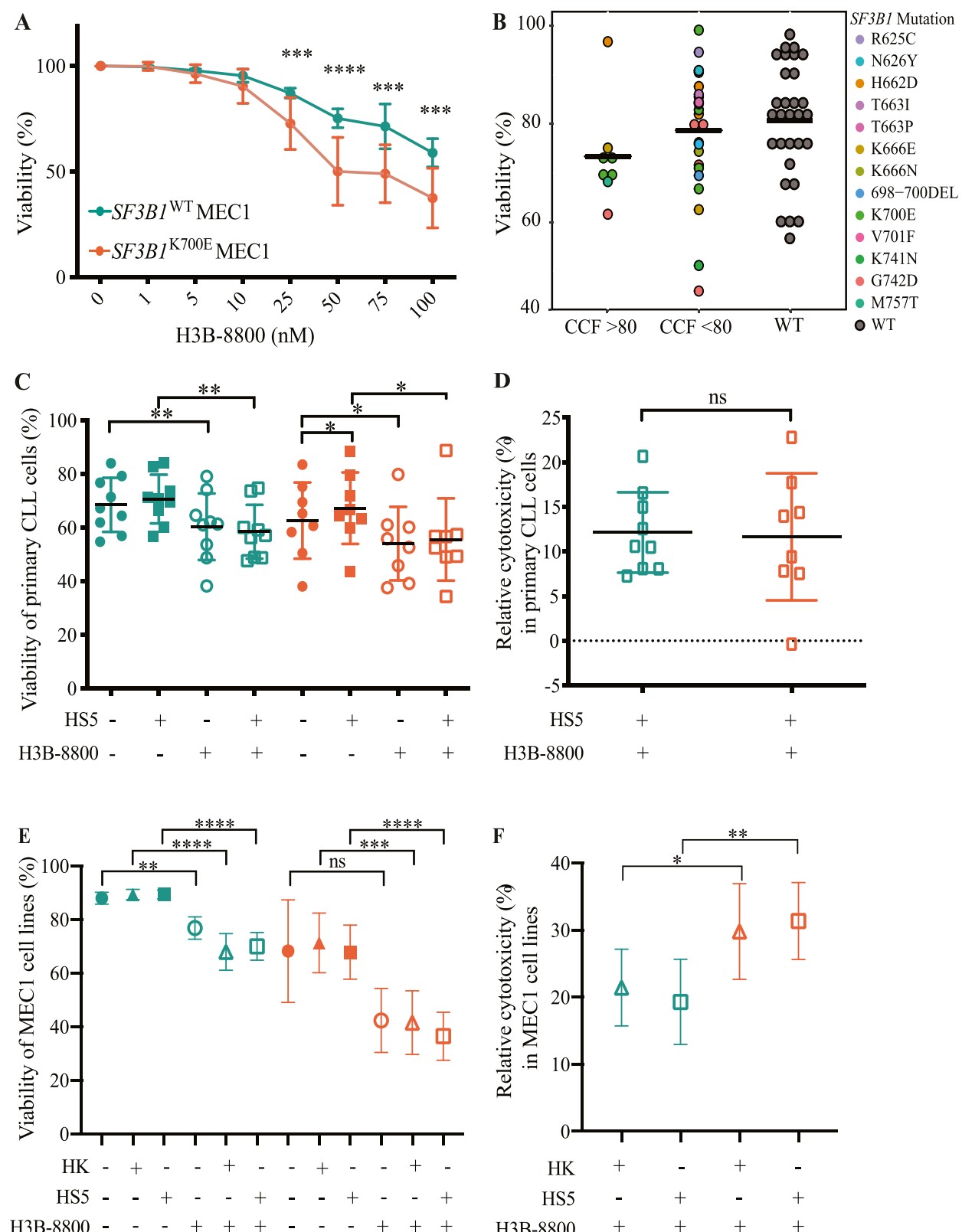

**Figure 6. Effects of H3B-8800 in splicing factor 3B subunit 1 (*SF3B1*)-mutated chronic lymphocytic leukemia (CLL) primary samples and cell lines.**
**(A)** Analysis of viability of CLL cell lines after 48 h treatment with the indicated concentrations of H3B-8800, measured by Annexin V and propidium iodide negativity by flow cytometry. *SF3B1*^WT refers to WT1, WT2, WT3, and *SF3B1*^K700 refers to MUT1, MUT2, MUT3. Replicates from four consecutive experiments are shown. The statistical significance of the comparison between the viability of *SF3B1*^WT and *SF3B1*^K700E cells was assessed using Mann–Whitney test (***P-value < 0.001, ****P-value < 0.0001).

3'ss in *ZNF561* in SF3B1$^{K700E}$ cells correlates with the use of a more downstream branch point, thus expanding the range of mechanisms involved. In contrast, in the case of *MAP3K7* or *TARBP1*, our results are more compatible with the same branch point being used in *SF3B1*$^{WT}$ and *SF3B1*$^{K700E}$ cells, which however lead to different relative use of the canonical and the upstream Alt3'ss. We therefore conclude that a variety of molecular mechanisms are involved in the effects of the K700E mutation on the recognition of branch points and/or 3'ss, which are normally proofread by the WT protein to avoid expression of abnormal isoforms.

Enrichment analyses revealed that the genes affected by these splicing alterations are involved in diverse biological processes, including immunological synapse formation, which is essential for the interaction of CLL cells with the microenvironment (Arruga et al, 2020); phosphatidylinositol biosynthetic processes, which are key components of the BCR-signaling pathway (Okkenhaug & Vanhaesebroeck, 2003); activation of MAPKK activity, involved in cell proliferation, transformation, and apoptosis (Wei & Liu, 2002; Yue & López, 2020); chromatin organization, in line with the recent finding of hypomethylated regions in *SF3B1*-mutated patient samples (Pacholewska et al, 2021); cell cycle checkpoint and negative regulation of apoptotic-signaling pathways, which can be crucial for cancer cell survival. The variety of pathways involved offers potential hints to elucidate the pathogenicity of *SF3B1* mutations in CLL, which is likely to be contributed by more than one of them. Among the genes differentially spliced in *SF3B1*-mutated samples, we observed several genes related to the activation of MAPKK, and in particular MAP3K7. MAP3K7 is a kinase that mediates tumor necrosis factor $\alpha$, interleukin-1$\beta$, and Toll-like receptor signaling through the NF-$\kappa$B, JNK, and MAPK pathways (Sato et al, 2005). We identified an alternative 3'ss located upstream of exon 5, which results in the degradation of the corresponding transcript by NMD (Li et al, 2021). We confirmed decreased MAP3K7 protein expression in the *SF3B1*$^{K700E}$ lines compared with *SF3B1*$^{WT}$ lines accompanied by increased NF-$\kappa$B activation in *SF3B1*$^{K700E}$ cases, as reported in other models (Liu et al, 2021). Other AS events that may confer advantages to *SF3B1*-mutated CLL cells have been described, including an Alt3'ss in *DVL2*, a gene that negatively regulates Notch pathway (Collu et al, 2012) and which has been linked to Notch1 activation in *SF3B1*-mutated CLL (Wang et al, 2016; Pozzo et al, 2020). The Notch pathway is involved in B-cell development (Arruga et al, 2018) and is activated in CLL patients harboring *NOTCH1* mutations (Puente et al, 2011; Arruga et al, 2014), conferring them specific biological features and being associated with poorer outcomes (Rossi et al, 2013; Villamor et al, 2013). Decreased methylation levels in nearby telomeric regions enriched in gene bodies of *NOTCH1* may contribute to NOTCH1 activation in *SF3B1*-mutated patients (Pacholewska et al, 2021). Another reported mechanism involves activation of an Alt3'ss in PPP2R5A, a protein phosphatase 2A subunit that promotes MYC stability and impairs apoptosis by increasing MYC S$^{62}$ and BCL2 S$^{70}$ phosphorylation in *SF3B1*-mutated tumors, including CLL (Liu et al, 2020).

We also found a cryptic isoform in the gene *ZDHHC16* that might be used as a surrogate marker for the functional impact of *SF3B1* mutations. It is detected in *SF3B1*-mutant samples with a CCF > 12%, exceptionally in cases with lower CCF and not in *SF3B1* WT or in cases with E862K or M757T mutations. Interestingly, in a phase I clinical trial using H3B-8800 to treat myeloid malignancies (NCT02841540), aberrant expression of an alternatively spliced isoform of *TMEM14C* identified patients with good responses (Steensma et al, 2021). However, it is important to point out that mutations (such as E862K or M757T) which do not reproduce alternative splicing alterations associated with hotspot mutations, may remain pathogenic through other mechanisms, for example, through functions of *SF3B1* on sister chromatid cohesion/mitotic chromosome segregation (Sundaramoorthy et al, 2014) or on nucleosome binding (Kfir et al, 2015).

The prognosis of CLL patients has significantly improved in the last decades with the introduction of new targeted therapies such as Bruton's tyrosine kinase inhibitors, phosphatidylinositol 3-kinase inhibitors, and BH3-only mimetics. Patients with *SF3B1* mutations, however, display worse prognosis (Byrd et al, 2019; Roberts et al, 2019), emphasizing the need for novel therapeutic options. SF3B1-binding splicing modulators, such as spliceostatin A, sudemycins and pladienolide B and its derivative FD-895, have shown stronger cytotoxic effects on CLL samples than on normal B lymphocytes (Kashyap et al, 2015; Xargay-Torrent et al, 2015; Larrayoz et al, 2016). A clinical trial using pladienolide B derivative E7107 for the treatment of solid tumors was withdrawn because of adverse ocular effects (Eskens et al, 2013; Hong et al, 2014). We have analyzed the effect of H3B-8800, a new orally bioavailable macrocyclic lactone small molecule that binds to the SF3B complex and modulates splicing (Seiler et al, 2018b), which has undergone a phase I clinical trial in patients with myeloid malignancies carrying mutations in spliceosome components, showing a promising safety profile and reduced red blood cell transfusion dependency (Steensma et al, 2021).

In preclinical models, including xenograft leukemia models with or without core spliceosome mutations, H3B-8800 has broad antitumor activity (Seiler et al, 2018b). H3B-8800 is cytotoxic for both WT and mutated CLL cells, although it shows preferential cytotoxic effect on *SF3B1*-mutant cells with high CCF (>80%) and isogenic *SF3B1*$^{K700E}$ MEC1 cell lines as compared with WT cells. The cytotoxic effect of H3B-8800 was observed even in the presence of protective stroma. Other splicing inhibitors such as sudemycin D6 or E7107 also display stronger cytotoxic effects on splicing factor–mutant cancer cells, including mutations in *SF3B1* and other

---

**(B)** Analysis of relative viability of primary CLL cells after 48 h treatment with 75 nM H3B-8800 measured by Annexin V and propidium iodide negativity by flow cytometry. *SF3B1*-mutated samples are divided into two groups according to their cancer cell fraction. The identity of the mutations is indicated by the color code. The crossbar represents the mean value. **(C, D)** Viability (C) and relative cytotoxicity (D) of primary CLL cells after treatment with 50 nM H3B-8800 for 48 h under co-culture conditions with HS-5 (bone marrow stromal cell line) cells. Green symbols correspond to *SF3B1*$^{WT}$ samples, orange symbols to *SF3B1*-mutated samples across different conditions specified in the x axis. Statistical significance assessment using Wilcoxon tests (*$P$-value < 0.05, **$P$-value < 0.01). **(E, F)** Viability of cell lines (E) and relative cytotoxicity (F) after treatment with 75 nM H3B-8800 for 48 h under co-culture conditions. Data information: HK: follicular dendritic cell line. HS-5: bone marrow stromal cell line. Green symbols correspond to *SF3B1*$^{WT}$ samples, orange symbols to *SF3B1*-mutated samples across different conditions specified in the x axis. Statistical significance assessment using Mann–Whitney test: **$P$-value < 0.01, ***$P$-value < 0.001, ****$P$-value < 0.0001, ns, non-significant.

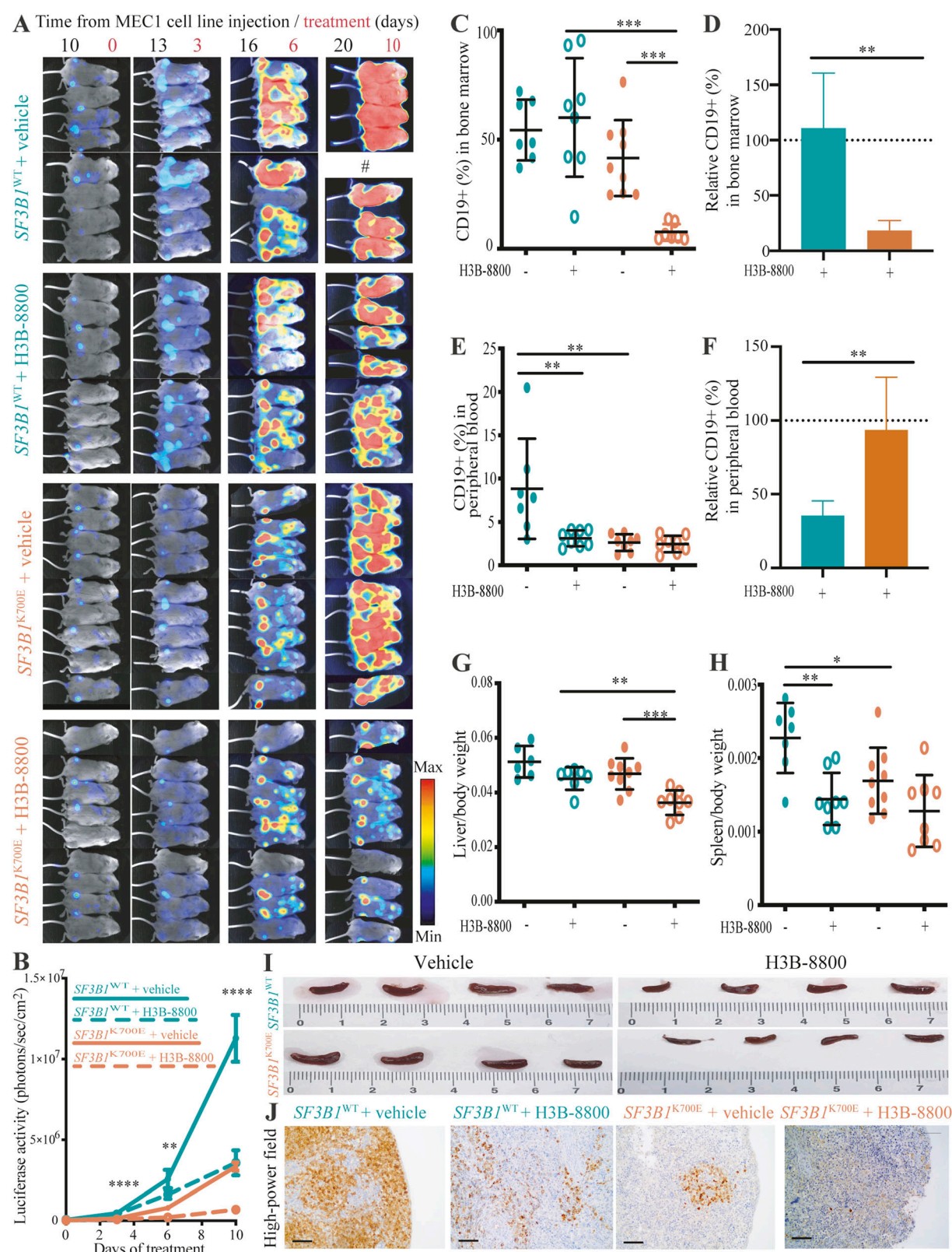

**Figure 7. Leukemic infiltration analysis of splicing factor 3B subunit 1 (*SF3B1*) WT and K700E chronic lymphocytic leukemia isogenic lines upon xenograft implantation in NSG mice and treatment with vehicle or H3B-8800 at 6 mg/kg.**
**(A)** Luciferase activity of *SF3B1*[WT] and *SF3B1*[K700E] MEC1 chronic lymphocytic leukemia isogenic lines xenografts in mice treated with vehicle or H3B-8800 for the times indicated, captured using AEQUORIA fluorescence/luminescence imaging system. # replaces the mouse which died before the experimental end point.

splicing factors such as *U2AF1* (Shirai et al, 2017) and *SRSF2* (Lee et al, 2016; Obeng et al, 2016). In CLL cells, sudemycin D1 and D6 were also more cytotoxic for cells bearing *SF3B1* mutations or other RNA splicing–related genes, both in vitro and in vivo (Xargay-Torrent et al, 2015), but these drugs have not entered into clinical trials.

H3B-8800 showed therapeutic effects in NSG mice bearing xenografts of *SF3B1*$^{WT}$ and *SF3B1*$^{K700E}$ MEC1 isogenic cell lines, as previously reported for other models harboring *SF3B1*$^{K700E}$ leukemic cells or *SRSF2*$^{P95H}$ chronic myelomonocytic leukemia cells (Seiler et al, 2018b). H3B-8800–treated *SF3B1*$^{K700E}$ group showed the lowest bone marrow and spleen infiltration after H3B-8800 treatment, arguing that H3B-8800 displays preferential lethality towards infiltrating *SF3B1*-mutant CLL cells. These findings would align with the idea that cancer cells harboring heterozygous mutations in *SF3B1* (or in other splicing factors such as *SRSF2* or *U2AF1*) are dependent on their WT allele for splicing function and cell survival (Zhou et al, 2015; Fei et al, 2016; Lee et al, 2016) and are therefore more sensitive to splicing inhibitors. Several lines of evidence support this idea. Mutations in the splicing machinery are mutually exclusive in myeloid malignancies (Yoshida et al, 2011). Accordingly, mutations in U1 snRNP, which are present in 10% of CLL patients, are mutually exclusive with *SF3B1* mutations (Shuai et al, 2019). This suggests that the splicing burden of splicing factor–mutant cells is close to compromising viability such that accumulating two perturbations makes cancer cells non-viable. Similarly, simultaneous expression of *SF3B1* and *SRSF2* mutations cause synthetic lethality (Lee et al, 2018), and single-cell analysis revealed that in rare cases in which mutations in splicing factors appear concomitantly, they are not hotspot mutations and display more limited effects on RNA splicing or RNA binding (Taylor et al, 2020). Considering these observations, allelic load may be the key factor for cell survival and tumor progression, as CLL tumors with low percentages of *SF3B1* mutation would be less sensitive to H3B-8800 treatment than tumors with high CCF.

Combined therapies involving SF3B1-binding splicing modulators and drugs used for CLL treatment have also been tested. For instance, the combination of sudemycin D1 with ibrutinib produces enhanced in vitro cytotoxicity involving IBTK's (BTK physiological inhibitor) splicing modulation (Xargay-Torrent et al, 2015). Also, combinations of spliceostatin A and BCL2 inhibitors (venetoclax or ABT-263) overcome the pro-survival effect of the microenvironment observed in monotherapy (Larrayoz et al, 2016). In addition, splicing modulation with the pladienolide B derivative E7107 sensitizes CLL cells to venetoclax therapy by reducing their dependency on MCL1 and increasing their dependence on BCL2 (Hacken et al, 2018). Furthermore, E7107 overcomes venetoclax

resistance in vivo in the Eμ-TCL1–based adoptive transfer murine model of CLL and increases mice overall survival alone or in combination with venetoclax (Hacken et al, 2018). Our initial results show that H3B-8800 also modulates MCL1 splicing and that H3B-8800 has a synergistic effect with venetoclax. A model has been proposed that SF3B-targeting splicing modulators preferentially perturb RNA splicing of MCL1 and BCL2A1, but not of BCL2L1 (BCLxL), leading to selective cytotoxicity in MCL1- or BCL2A1-dependent cancer cells. In combination with BCLxL/BCL2 inhibitors, splicing modulators can enhance cytotoxicity through broader inhibition of the BCL2 family genes that act cooperatively to promote anti-apoptosis/pro-survival in cancer cells (Aird et al, 2019).

Different SF3B-binding molecules display common and differential effects on splicing regulation (Vigevani et al, 2017). In general, they cause the retention of short introns with high GC content and the skipping of short exons harboring weak splice sites, as it has been observed with spliceostatin A, sudemycin, E7107 and H3B-8800, among others (Lee et al, 2016; Shirai et al, 2017; Vigevani et al, 2017; Seiler et al, 2018b). We find that exons skipped upon H3B-8800 treatment of CLL cells are shorter, flanked by shorter introns and exons, and featuring stronger 5'ss and 3'ss than alternative non-changing exons. Our results also show that H3B-8800 preferentially induces retention of introns characterized by short length, high GC content (as also reported for H3B-8800–treated K562 cell line [Seiler et al, 2018b]) and low differential GC content compared with the following exon. We observed that these introns feature weaker 5' and 3'ss and shorter flanking exons. Short, GC-rich introns have been shown to be recognized by intron-defined splicing mechanisms whose alteration is associated with intron retention (De Conti et al, 2013). Altogether, these findings suggest that H3B-8800 selectively modulates introns and exons with specific features. The selective mechanism of action of H3B-8800, in combination with its unique pharmacological profile compared with previously described splicing inhibitors, supports clinical testing of H3B-8800 in genetically defined subsets of tumors harboring RNA splicing factor mutations.

In summary, the work presented here reports (1) the first generation of isogenic genome–edited CLL cell lines differing only by the presence or absence of a single amino acid substitution in SF3B1, (2) the identification of a wider variety of 3' splice site activation events upon *SF3B1* mutation compared with previous reports, (3) provides evidence that the SF3B1 inhibitor H3B-8800 displays therapeutic effects in mouse models of CLL, particularly when used on CLL cells harboring *SF3B1* mutations, and (4) provides evidence of synergism between a SF3B1-targeting drug and venetoclax.

---

(A, B) Quantification of the results in panel (A). The color and line codes indicate the different experimental conditions and treatments. (C, D, E, F) Analysis of leukemic infiltration in the bone marrow (C, D) and peripheral blood (E, F), measured as the percentage of CD19$^+$ cells in flow cytometry assays (C, E) for *SF3B1*$^{WT}$ and *SF3B1*$^{K700E}$ xenograft-implanted NSG mice, treated with vehicle or H3B-8800, as indicated, or relative to treatment with vehicle (D, F). *SF3B1*$^{WT}$ samples are indicated in green, *SF3B1*$^{K700E}$ samples in orange. (G, H) Normalized quantification of the liver (G) and spleen (H) weights from mice harboring *SF3B1*$^{WT}$ (green) or *SF3B1*$^{K700E}$ (orange) xenografts treated with vehicle or H3B-8800 as indicated. (I) Representative spleens from NSG mice implanted with *SF3B1*$^{WT}$ and *SF3B1*$^{K700E}$ MEC1 cell lines after H3B-8800 or vehicle administration, as indicated. (J) CD79a immunohistochemical staining of spleens from H3B-8800–treated versus vehicle-treated mice implanted with *SF3B1*$^{WT}$ and *SF3B1*$^{K700E}$ MEC1 cell lines as indicated. The scale bar indicates 75 nm. (B, C, D, E, F, G, H) Data information: statistical significance was assessed using mixed effect analysis (B) and Mann–Whitney test (C, D, E, F, G, H) (*$P$-value < 0.05, **$P$-value < 0.01; ***$P$-value < 0.001, ****$P$-value < 0.0001).

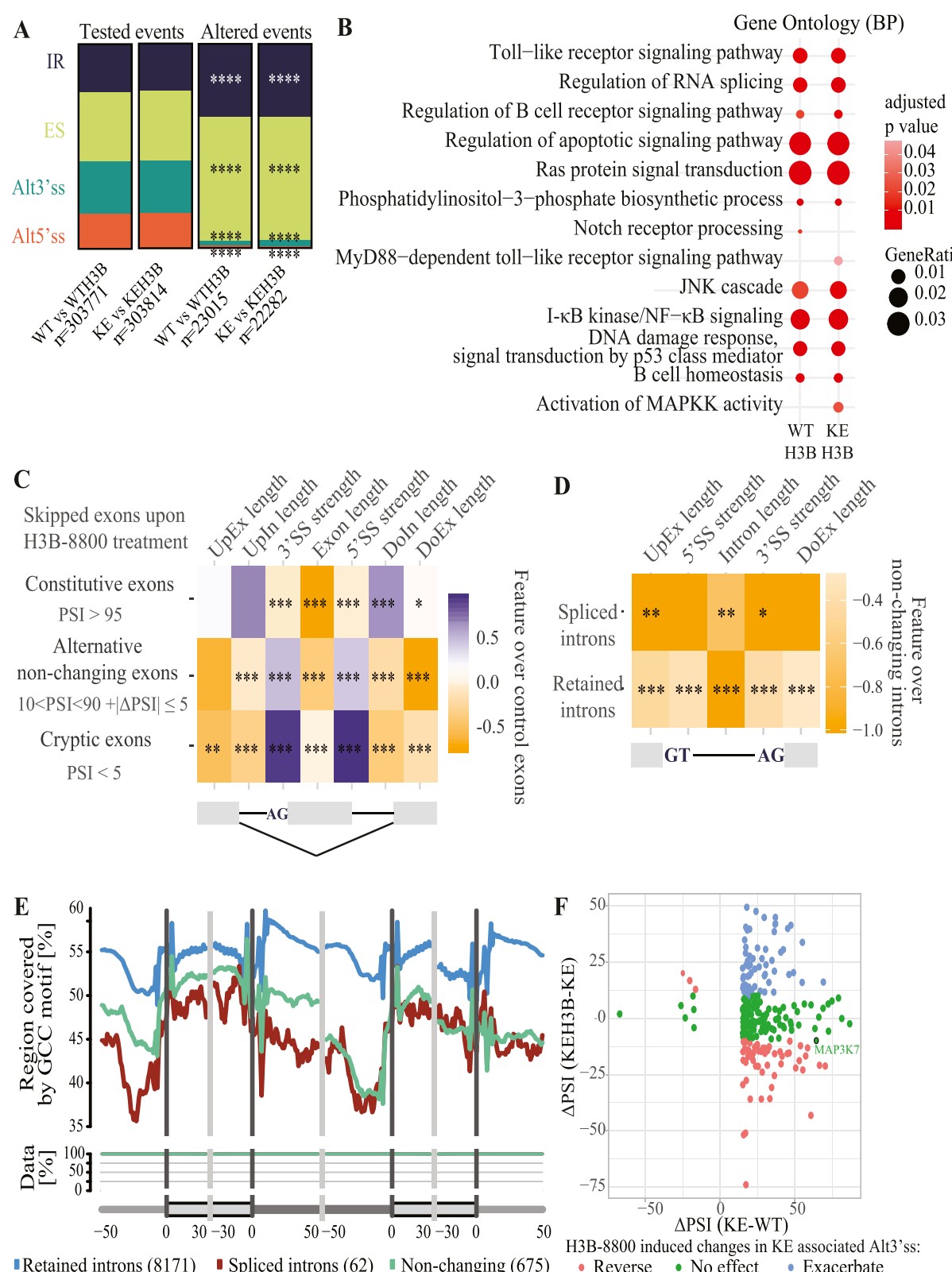

**Figure 8. H3B-8800 induces intron retention and exon skipping events in H3B-8800–treated *SF3B1*WT and *SF3B1*K700E MEC1 cell lines.**
**(A)** Relative frequency of differentially spliced (|ΔPSI| ≥15) AS event types in *SF3B1*WT and *SF3B1*K700E MEC1 cell lines upon H3B-8800 treatment compared with all assessed AS events using *vast-tools* for each comparison (WT versus WTH3B: *SF3B1*WT versus *SF3B1*WT treated with H3B-8800; KE versus KEH3B: *SF3B1*K700E versus *SF3B1*K700E treated with H3B-8800). Test of equal proportion (****$P < 2.2 \times 10^{-16}$). **(B)** Gene Ontology enrichment analysis of biological processes for all the AS differentially expressed

# Materials and Methods

## CLL samples and cell lines

PBMCs from CLL patients (with ≥90% tumor B cells) and healthy donors were isolated, cryopreserved, and stored within the hematopathology collection registered at the Biobank (R121004-094; Hospital Clinic-IDIBAPS). Clinical and biological data of each patient are detailed in Table S1. Written informed consent was obtained from all patients. The study was approved by the Hospital Clínic of Barcelona Ethics Committee. *SF3B1* mutations in PBMCs were analyzed in previous sequencing studies (Puente et al, 2011, 2015; Quesada et al, 2013). MEC1 (ACC497; DSMZ) CLL cell lines were cultured at $5 \times 10^5$ cells/ml in IMDM with GlutaMAX (GIBCO) supplemented with 10% (vol/vol) heat-inactivated FBS and 1% (vol/vol) penicillin–streptomycin 10,000 U/ml (GIBCO) in a humidified atmosphere at 37°C containing 5% carbon dioxide. The follicular dendritic cell line HK, kindly provided by Dr. Yong Sung Choi (Alton Ochsner Medical Foundation) (Kim et al, 1994), was cultured in RPMI1640 (GIBCO) supplemented with 20% FBS, 1% (vol/vol) penicillin–streptomycin 10,000 U/ml, and 1% (vol/vol) GlutaMax 100X. HS-5 (CRL-11882; ATCC) cell line was cultured in DMEM (GIBCO) supplemented with 10% (vol/vol) heat-inactivated FBS, 1% (vol/vol) penicillin–streptomycin 10,000 U/ml, and 1% (vol/vol) GlutaMax 100X. Every batch of cell lines was tested for mycoplasma with EZ-PCR Mycoplasma Detection Kit (Biological Industries) periodically when they were in culture. Identification of all cell lines was carried out using the GenePrint 10 system (Promega).

## Generation and characterization of CRISPR Cas9–edited cell lines

ssODN complementary to the non-target strand (Richardson et al, 2016) with two nucleotides changes compared with the reference sequence was designed using DESKGEN AI library design software. We replaced c.2098A>G to induce a lysine to glutamic acid change and introduced a PAM silent mutation, changing XGG PAM sequence to XGA (c.2106G>A) to avoid further recognition of the PAM sequence once it was properly repaired (Paquet et al, 2016). Oligonucleotides used for the generation and characterization of *SF3B1*[K700E] MEC1 cell line are shown in Table S14. Annealing of crRNA and Alt-R CRISPR–Cas9 tracrRNA, ATTO 550 (IDT) was carried out at 95°C for 5 min, then the assembly of the crRNA:tracrRNA duplex (10 *μ*M) and the Cas9 protein (10 *μ*M) was performed and mixed with ssODN (10 *μ*M) and Alt-R Cas9 Electroporation Enhancer (IDT). MEC1 cell lines were electroporated using the Neon Transfection System (Thermo Fisher Scientific) with a 20 ms pulse at 1,600 V and seeded. Serial dilution in single-cell cultures was performed for 3 wk and then clones were screened using Taq*α*1 restriction enzyme, recognizing the c.2106G>A from the PAM sequence (Fig 1B). 30,000 MEC1 cell/well were seeded in a 96-well flat bottom polystyrene culture plate and incubated for 48 h. DNA extraction was carried out using QIAamp DNA Micro Kit ($<1 \times 10^6$) or QIAamp DNA Mini Kit ($>1 \times 10^6$) (QIAGEN); 50 ng DNA was used to quantify *SF3B1* K700E mutation by digital PCR (Bio-Rad).

Cell viability was measured by flow cytometry using Annexin V (FITC) and propidium iodide (Invitrogen). For the MTT assay, 10,000 cells/well were seeded and incubated for 48 h. Then, thiazolyl blue tetrazolium bromide 98% at a concentration of 0.5 mg/ml in PBS was added to the cells and were incubated for 2 h until the presence of violet crystal was observed at the optical microscope (Leica DMI1). The stop solution (isopropanol and HCl 24:1) was added and the cells were carefully resuspended before reading the plates at the TECAN spectrophotometer at wavelengths of 570 and 670 nm.

## RNA-seq studies

RNA from CLL samples were collected before any treatment administration and were processed and quantified as described in the study by Puente et al (2015). RNA-seq libraries were prepared from total RNA using the TruSeq RNA Sample Prep Kit v2 (Illumina Inc.) with minor modifications, and each library was sequenced using TruSeq SBS Kitv3-HS on a HiSeq2000 (Illumina Inc.) as described previously (Puente et al, 2015). Four million *SF3B1*[WT] or *SF3B1*[K700E] MEC1 cell lines were treated with 75 nM H3B-8800 or DMSO for 6 h, and RNA extraction was performed using the RNeasy Mini kit (QIAGEN). Stranded mRNA–seq libraries were prepared and two samples/lane were sequenced, following 2 × 125 nt paired-ended protocol at the CRG Genomics Core Facility on a HiSeq 2500 v4 (Illumina Inc). Triplicates were sequenced for each condition in separate lines. All the samples included in the comparison

(|ΔPSI| ≥ 15 and percent spliced in [PSI] range < 5 upon H3B-8800 treatment) in *SF3B1*[WT] and *SF3B1*[K700E] MEC1 cell lines after treatment with 75 nM H3B-8800 for 6 h calculated using enrichGO. GO terms are indicated on the left, and the enrichment properties (adjusted *P*-value and gene ratio, the ratio of the affected genes in each pathway) indicated by the heatmap color and size of the circles, respectively. GO terms shown have an adjusted *P*-value < 0.05 and a *Q*-value < 0.05. **(C)** Sequence feature analysis of exons skipped upon H3B-8800 treatment of *SF3B1*[WT] MEC1 cells using Matt (Gohr & Irimia, 2019). Significant features are indicated at the top of the figure and the degree and statistical significance level for the comparison with constitutive, alternative non-changing, and cryptic exons indicated by the heatmap and Mann–Whitney test (*$P$ < 0.05, **$P$ < 0.01, ***$P$ < 0.001). 5'ss and 3'ss strength are calculated by the maximum entropy score of 3'ss using a model trained with human splice sites (Yeo & Burge, 2004). Exons skipped upon H3B-8800 treatment were defined as |ΔPSI| ≥ 15 and PSI range < 5 upon H3B-8800 treatment. Constitutive exons were defined as displaying PSI > 95 in both groups (n = 4,922); alternative non-changing exons as 10 < PSI < 90 in at least one group and |ΔPSI| ≤ 5 (n = 3,288) and cryptic exons as PSI < 5 in both groups (n = 4,976). **(D)** Sequence feature analysis of retained and more spliced introns upon H3B-8800 treatment of *SF3B1*[WT] MEC1 cells using *Matt* (Gohr & Irimia, 2019). 5'ss and 3'ss strength are calculated by the maximum entropy score of 3'ss using a model trained with human splice sites (Yeo & Burge, 2004). Retained introns were defined as |ΔPSI (control—treated)| ≥ 15 and PSI range < 5, spliced introns as ΔPSI ≤ 15 and PSI range < 5 and non-changing introns as −1 < ΔPSI < 1, 5 < PSI < 95 in *SF3B1*[WT] control group. Significant features are indicated at the top of the figure, and the degree and statistical significance level for the comparison of retained and more spliced introns with non-changing introns indicated by the heatmap and Mann–Whitney test (*$P$ < 0.05, **$P$ < 0.01, ***$P$ < 0.001). **(E)** Distribution of GC content along the genomic exon/intron region (RNA map) for retained, more spliced, and non-changing introns using Matt (Gohr & Irimia, 2019). The upper plot shows the frequency of GC content motif; the middle plot shows the coverage of the region, and the lower scheme, the length (in nucleotides) of the regions of exons and introns considered for the analysis. **(F)** Effect of H3B-8800 on splicing changes induced by *SF3B1*[K700E] mutation. The similarity between *SF3B1* mutation and H3B-8800 treatment induced ΔPSI direction is shown in color code. |ΔPSI| ≤ 10 was set as a cut of to define the events affected less by H3B-8800 treatment. Alt3'ss event in MAP3K7 (studied in Fig 3) is highlighted. Data information: KE: *SF3B1*[K700E]; KEH3B: *SF3B1*[K700E] treated with H3B-8800, and WT: *SF3B1*[WT].

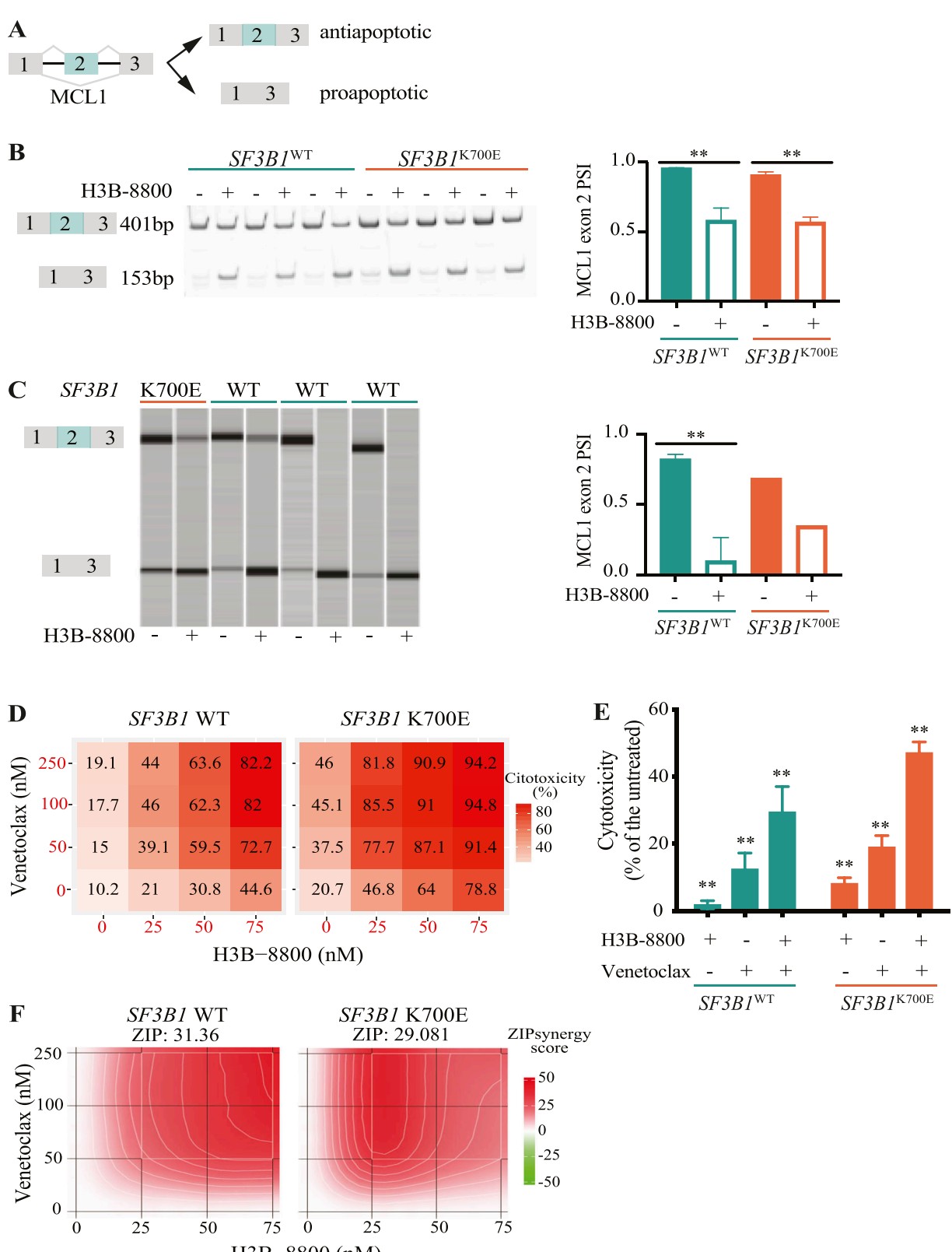

**Figure 9. H3B-8800 modulates MCL1 AS and displays synergistic effects with venetoclax in MEC1 chronic lymphocytic leukemia (CLL) cell lines.**
**(A)** Schematic representation of the AS event involving MCL1 exon 2. **(B)** Validation of MCL1 exon 2 skipping in $SF3B1^{WT}$ and $SF3B1^{K700E}$ isogenic MEC1 cell lines after 6 h treatment with 75 nM H3B-8800. RT–PCR amplification products corresponding to $SF3B1^{WT}$ and $SF3B1^{K700E}$ isogenic MEC1 cell lines treated with vehicle or H3B-8800, using primers complementary to exons 1 and 3, were analyzed by polyacrylamide gel. Right panel: quantification of exon 2 percent spliced in values under the different

passed the FastQC quality control from *MultiQC* (Ewels et al, 2016).

## Splicing analyses

Splicing analyses were carried out using the toolset Vertebrate Alternative Splicing and Transcription Tools (Irimia et al, 2014; Tapial et al, 2017) (*vast-tools*, v2.5.1) (https://github.com/vastgroup/vast-tools). For each event, minimum 10 reads coverage was required.

225 *SF3B1* WT cases and 15 cases harboring *SF3B1* mutations (8 K700E, 3 H662D, 2 R625C, 1 L743F, and 1 T663I) with high CCF (≥50%) were analyzed, excluding samples with mutations E862K and M757T in *SF3B1*, in other splicing factors or in other RNA metabolism–related genes (*BUD13*, *CARNS1*, *CELF4*, *DDX3X*, *EIF4A3*, *MAGOH*, *MPHOSPH10*, *NXF1*, *RNF113A*, *SKIVV2L2*, *SF3B4*, *SRSF2*, *SRSF7*, *XPO1*, *U1*, *U2AF2*, *ZRSR2*). We made two comparisons considering the distribution of the patients in PC1 for all the AS events detected (Fig 2A) because of the concerns about the sampling process reported previously (Dvinge et al, 2014). We compared the samples with PC1 ≥0 on one side (8 *SF3B1*-mutated samples versus 136 samples *SF3B1*^WT samples) and the samples with PC1 < 0 (7 *SF3B1*-mutated samples versus 89 samples *SF3B1*^WT samples). AS events needed to have at least 10 reads in 5 of *SF3B1*-mutated samples and 20 of the *SF3B1*^WT samples to take them into consideration for the comparison and a |ΔPSI| higher than 15% and range of 5 between *SF3B1* mutated and *SF3B1*^WT samples. To detect differentially spliced events between two conditions (*SF3B1* mutation or H3B-8800 treatment), we required a |ΔPSI| of at least 15% between the merged replicates under each condition and a minimum range of 5 between the PSI values under the two conditions in MEC1 cell lines.

For a wider search of annotated and not annotated Alt3'ss, we developed *JuncExplorer* (https://gitlab.com/aghr/juncexplorer). *JuncExplorer* allows the identification and quantification of annotated and non-annotated (de-novo) splice junctions from RNA-seq reads mapped with a splice-aware mapper, for example, STAR (see the Materials and Methods section on Gene expression analysis). PSI values are calculated by converting the read numbers into relative frequencies per case for each of its splice sites for each of the 5' and 3'ss. Alt 3'ss cases were considered when a unique GT donor site (5') had several AG acceptor sites (3') within a single intron. To detect differentially spliced events between two conditions (*SF3B1* mutation or H3B-8800 treatment), we required a |ΔPSI| of at least 15% and range of 5 between the merged replicates under each condition. Cryptic 3'ss correspond to alternative 3'ss not detected in *SF3B1* WT samples and detected in *SF3B1*-mutated samples. Downstream analysis of AS events was performed using *Matt* (https://gitlab.com/aghr/matt) (Gohr & Irimia, 2019) with

special focus on exon and intron features, splice sites strengths (Yeo & Burge, 2004), and BP predictions (Corvelo et al, 2010).

For the sequence features analysis associated with AS changes induced by the drug in *SF3B1*^WT cells, *Matt* software (Gohr & Irimia, 2019) was used. The group of regulated exons (|ΔPSI (control and H3B-8800 treated)| ≥ 15 and PSI range < 5 upon H3B-8800 treatment) was compared with different control groups (constitutive exons: PSI > 95 in both groups [n = 4,922]; alternative non-changing exons in at least one group: 10 < PSI < 90 and |ΔPSI| ≤ 5 [n = 3,288]; and cryptic exons: PSI <5 in both groups [n = 4,976] Fig 8C). The retained introns upon H3B-8800 treatment were compared with introns that were spliced more efficiently and with introns that did not change (−1 < ΔPSI < 1, 5 < PSI < 95 in *SF3B1*^WT control group) upon drug treatment.

Prediction of impact on protein sequence was obtained from VastDB version 3 (https://vastdb.crg.eu/wiki/Downloads) (Irimia et al, 2014). First, AS sequences were mapped to non-coding (UTRs) or coding sequences (CDS). Then, the events were predicted to disrupt the ORF if their inclusion or skipping would induce a frameshift; they would induce a premature stop codon predicted to trigger NMD or an in-frame stop codon would generate a protein 100 amino acids shorter than the reference protein. The rest of the CDS mapping events were considered to preserve the transcript's coding capability.

Pearson coefficient correlation plots were obtained with *ggplot* and *ggpubr* packages in R (v3.6.3).

## Gene expression analysis

Illumina universal adapters sequences were removed from the RNA-Seq reads with *Cutadapt* version 2.4 discarding all reads with length shorter than 15. The adapter-trimmed reads were mapped with *STAR* 2.7.1.a (Dobin et al, 2013) to the human genome version GRCh38. Read counts per gene were generated with *STAR* (–quantMode GeneCounts) using uniquely mapping reads only. DESeq2 version 1.14.1 (Love et al, 2014) was used to determine fold changes and *P*-values for all annotated genes. Data from CLL samples were corrected for batch effects, caused by the different processing time of samples. Therefore, we define a batch variable and added it to the model definition of the generalized linear model used by *DESeq2*. Heatmaps were carried out using *iDEP* v.0.94 and v.0.96 (Ge et al, 2018).

Differentially expressed genes or genes bearing differentially spliced (|ΔPSI| ≥ 15 and range of 5) AS events (Alt3'ss, Alt5'ss, IR, and ES) between *SF3B1* WT and mutated patients' and MEC1 cell line's groups were analyzed using *enrichGO* and *clusterProfiler* packages in R (v3.6.3). Statistical significance was defined with

---

conditions (mean and SD). Statistical significance was assessed using unpaired *t* tests (\*\**P*-value < 0.01; \*\*\**P*-value < 0.001). **(C)** Validation of MCL1 exon 2 skipping in four patient samples after 3 h of treatment with 75 nM H3B-8800, analyzed by RT–PCR as in (B) and capillary electrophoresis. Right panel: quantification of exon 2 percent spliced in values under the different conditions (mean and SD). Statistical significance was assessed using unpaired *t* tests (\*\**P*-value < 0.01; \*\*\**P*-value < 0.001). **(D)** Dose-response cytotoxicity matrices for the indicated doses of H3B-8800 and venetoclax combination therapy at 48 h in *SF3B1*^WT and *SF3B1*^K700E MEC1 cell lines. Cytotoxicity was assessed by Annexin V and propidium iodide positivity by flow cytometry. **(E)** Cytotoxicity after incubation for 48 h with 100 nM venetoclax and/or 50 nM H3B-8800 in *SF3B1*^WT and *SF3B1*^K700E MEC1 cell lines. Cytotoxicity was assessed by Annexin V and propidium iodide positivity by flow cytometry. Statistical significance was assessed using Wilcoxon signed rank test (\*\**P*-value < 0.01). **(D, F)** Zero interaction potency synergy score representation based on the dose-response cytotoxicity matrices from panel (D) for the indicated H3B-8800 and venetoclax combination doses in *SF3B1*^WT and *SF3B1*^K700E MEC1 cell lines. Synergy scores were calculated using *SynergyFinder* (Ianevski et al, 2020).

adjusted *P*-value < 0.05 and the positive false discovery rate with *Q*-value < 0.05.

## Experimental validation of RNA-seq data

RNA extraction and DNAse treatment in *SF3B1*^WT and *SF3B1*^K700E MEC1 cell lines were performed using Maxwell simplyRNA kit (Promega), Illustra RNAspin Mini RNA Isolation Kit (GE Healthcare), or RNeasy Mini kit (QIAGEN), following the manufacturers' instructions. 50 ng of total RNA were reverse transcribed with SuperScript III (Invitrogen), and PCR reactions were carried out using GoTaq enzyme (Promega). PCR products were separated by electrophoresis on 6% acrylamide gels, analyzed with GelDocTM XR+ (Bio-Rad) and quantified with *ImageJ* v1.53a. Primers used are listed in Table S15. TARBP1 PCR products were analyzed by fragment analysis in an ABI 3130 Genetic Analyzer (Thermo Fisher Scientific) using *GeneMapper* software v6.0 (Thermo Fisher Scientific).

Total RNA from CLL primary samples was extracted using the TRIzol reagent (Invitrogen), and 1 *µ*g of total RNA was then reverse transcribed to cDNA. Aberrant and canonical isoforms of MAP3K7 and ZDHHC16 were quantified in StepOnePlus Real-Time PCR System (Applied Biosystems) using specific primers (Table S16). The relative expression of each gene was analyzed by the comparative cycle threshold (Ct) method (ΔΔCt) using *RPLP0* as an endogenous housekeeping gene. mRNA expression levels were provided as arbitrary quantitative PCR units, taking as a calibrator the median of the WT samples. In the case of ZDHHC16 cryptic isoform, which was not detected in the WT samples, the median of the *SF3B1*-mutated samples was taken as reference.

## Minigenes

pCMV 57 Δi AdML plasmid harboring two exons and an intron from the adenovirus major late promoter transcript, flanked by PT1 and PT2 sequences that allow detection of transcripts from the expression vector (Sakamoto et al, 1992) by RT–PCR analysis, was used as a scaffold in which different 3'ss (from 20 nucleotides upstream of the predicted BP to five nucleotides downstream of the 3'ss) were introduced replacing equivalent sequences of the AdML intron (Table S11). For this, pCMV 57 Δi AdML plasmid was amplified by PCR using TaqPlus Precision (Agilent), deleting the AdML intron two sequences and DNA oligonucleotide primers corresponding to the sequences of interest were ordered to IDT and amplified with GoTag (Promega). The PCR products were purified by QIAquick PCR Purification Kit (QIAGEN), and the methylated DNA template was digested with the enzyme DpnI (New England BioLabs). The amplified vector was purified by agarose gel electrophoresis and purified using QIAquick Gel Extraction Kit (QIAGEN) following the manufacturer's instructions. 20 ng of the vector were used for Gibson cloning at a 1:8 vector-insert ratio with a mix of reagents from the CRG Protein Technologies Unit core facility. Stellar bacteria (Takara Bio) were used for transformation. Single colonies from LB ampicillin plates were grown overnight in LB supplemented with ampicillin, DNA was purified using QIAprep Spin Miniprep Kit (QIAGEN), and minigenes were sequenced (Eurofins) to verify the products of mutagenesis.

MEC1 *SF3B1*^WT and *SF3B1*^K700E cell lines were transfected with 50 ng of minigene DNA plasmids and Alt-R Cas9 Electroporation Enhancer (IDT) using Neon Transfection System (Thermo Fisher Scientific) with a 20 ms pulse at 1,600 V. 24 h later, cells were collected for RNA extraction and validated by RT–PCR. The PCR products from the *TARBP1* gene were analyzed by fragment analysis as described above. PCR products of other minigenes were separated by electrophoresis on standard native polyacrylamide gels as described above.

## Western blots

Cells (5 × 10^6) were lysed for 30 min in Tris–Triton-X Lysis buffer (20 mM Tris–HCl, pH 7.6, 150 mM NaCl, 1% Triton X-100) supplemented with protease and phosphatase inhibitors (Sigma-Aldrich). Lysates were centrifuged at 15,000*g* for 15 min at 4°C, and protein concentration assessed by Bradford assay (Bio-Rad). Total extracts were separated by 10% SDS–PAGE and transferred to an Immobilon-P PVDF membrane (Merck Millipore). Primary and secondary antibodies used for Western blot analyses are shown in Table S17. Chemiluminescence was detected by using ECL substrate (Thermo Fisher Scientific) or Amersham ECL Select Western Blotting Detection Reagent (Cytiva) on a mini-LAS4000 Fujifilm device (GE Healthcare) and quantified by Multi Gauge software (FUJIFILM).

## NF-κB activity measurements

NF-κB activity was measured using ELISA-based TransAM NF-κB Family Transcription Factor Assay Kit (Active motif) in nuclear extracts obtained using nuclear extract kit as recommended by the manufacturers.

## Cytotoxicity studies

30,000 MEC1 cell/well or 200,000 primary cells/well were seeded in a 96-well flat bottom polystyrene culture plates (Thermo Fisher Scientific) and incubated at different concentrations for 48 h with H3B-8800 (kindly provided by H3B Biomedicine) and/or venetoclax (ABT-199; Selleckchem). Cytotoxicity was measured by Annexin V (Pacific blue/FITC) and PI-mediated (Invitrogen) and flow cytometry (Attune Acoustic Focusing Cytometer from Applied Biosystems for primary cells and BD LSRFortessa Flow Cytometer for the cell lines). For co-culture experiments, HS-5 and HK cells were seeded on day 1 at a density of 20,000 cell/well and 10,000 cells/well, respectively. On day 2, 20,000 MEC1 cells/well or 200,000 primary cell/well were added and incubated with 50–75 nM H3B-8800 for 48 h. Cytotoxicity was assessed by flow cytometry as described above, after selection of MEC1 cells by CD19^+ (BD) positivity. Synergism between H3B-8800 and venetoclax was calculated using *SynergyFinder* 2.0 (Ianevski et al, 2020).

## Stable expression of luciferase and GFP MEC1 cell lines

HEK 293T cells were transfected using jetPEI with pLV430G-oFL-T2A-eGFP, a lentivirus expressing the luciferase gene cloned in an EGFP-expressing vector (gift from Amer Najjar, M. D. Anderson Cancer Center), and viral stocks were collected from the culture supernatant 48 h post-transfection. MEC1 cell lines were infected, and GFP-positive cells were sorted by flow cytometry 7 d afterwards.

 **Life Science Alliance**

### In vivo model

10 millions of GFP–$SF3B1^{K700E}$ (in 17 mice) or GFP–$SF3B1^{K700K}$ (in 16 mice) MEC1 CLL cell lines were injected intravenously on 6–8 wk old NOD-SCID interleukin-2 receptor gamma ($IL2R\gamma)^{null}$ (NSG) mice. Leukemic infiltration was followed by bioluminescence using a fluorescence/luminescence imaging system AEQUORIA (Hamamatsu Photonics K.K.). 10 d after the injection, once leukemic infiltration was detected by bioluminiscence, H3B-8800 (6 mg/Kg) or vehicle control (95% methylcellulose, 0.5% and 5% ethanol 100%) was administrated orally and daily for 10 d until the mice were euthanized 24 h after receiving the last dose. The experimental design was approved by the Animal Testing Ethic Committee of the University of Barcelona and Generalitat de Catalunya (#9680; Protocol). Peripheral blood was extracted from the submaxillary vein. Bone marrow was obtained from femurs by cutting the neck and next to the condyles, and then injecting IMDM with a syringe and a 25G needle to facilitate bone marrow extraction. Peripheral blood, bone marrow, spleen, and liver samples were first processed with ACK lysing buffer (Lonza), and then, leukemic infiltration was measured using CD19⁺ (Invitrogen) to detect the tumoral B lymphocytes in a BD LSRFortessa Flow Cytometer. Mouse spleens were fixed in 10% formaldehyde, dehydrated by ethanol after 24 h, and embedded in dissolved paraffin. The specimens were finally stained by hematoxylin–eosin (HE) and CD79a (Roche) immunohistochemical staining and then analyzed in an Olympus BX53 optical microscope.

### Statistical analysis

Statistical tests were performed as indicated in figure legends with R (v3.6.3) or GraphPad Prism (v8.4.0).

## Data Availability

RNA-seq data from $SF3B1^{WT}$ and $SF3B1^{K700E}$ MEC1 cell lines treated with H3B-8800 are available in the following database: Gene Expression Omnibus GSE190087. RNA-seq data from patient samples were accessed from the previously available (Puente et al, 2015) database: European Genome-Phenome Archive (EGA, http://www.ebi.ac.uk/ega/), which is hosted at the European Bioinformatics Institute (EBI), under accession number EGAS00000000092.

## Supplementary Information

## Acknowledgements

I López-Oreja is a recipient of a PhD4MD Program Fellowship (CRG-IDIBAPS). H Playa-Albinyana is a recipient of a predoctoral fellowship from the Spanish Ministry of Universities (FPU19/03110). The work of the authors is supported by the Spanish Ministry of Science and Innovation through the Plan Estatal de Investigación Científica y Técnica y de Innovación (RTI2018-094584-B-I00; PID2021-123165OB-I00; BFU-2017-89308-P, PID2020-114630GB-I00) and was cofounded by the European Regional Development Fund (ERDF), European Research Council (ERC) under the European Union's Horizon 2020 research and innovation program ERC-AdG-LS2-670146 l, CERCA program from Generalitat de Catalunya, Centre of Excellence Severo Ochoa Award (CEX2020-001049) to the Centre for Genomic Regulation of the Barcelona Institute of Science and Technology, Centro de Investigación Biomédica en Cáncer (CIBERONC) (CB16/12/334), Agència de Gestió d'Ajuts Universitaris i de Recerca, Generalitat de Catalunya (2017 SGR 1009; 2021 SGR-1294) and Asociación Española Contra el Cancer (AECC) (PRYGN223298VALC). We thank Elias Campo for sharing transcriptomics data from ICGC consortium and for reviewing the manuscript. We thank Suzanne Mays, Estefania Mancini, Malgorzata Rogalska, and Ferran Nadeu for their help and all the laboratory members from Juan Valcarcel's group.

## Author Contributions

I López-Oreja: conceptualization, data curation, formal analysis, validation, investigation, visualization, methodology, project administration, and writing—original draft, review, and editing.
A Gohr: conceptualization, software, formal analysis, and writing—original draft.
H Playa-Albinyana: validation, investigation, and visualization.
A Giró: formal analysis, validation, investigation, and methodology.
F Arenas: resources, formal analysis, investigation, and methodology.
M Higashi: data curation, formal analysis, validation, and investigation.
R Tripathi: formal analysis, validation, and investigation.
M López-Guerra: conceptualization and methodology.
M Irimia: conceptualization and software.
M Aymerich: resources.
J Valcárcel: conceptualization, resources, supervision, funding acquisition, methodology, project administration, and writing—original draft, review, and editing.
S Bonnal: conceptualization, formal analysis, validation, investigation, visualization, methodology, project administration, and writing—original draft, review, and editing.
D Colomer: conceptualization, resources, supervision, funding acquisition, project administration, and writing—original draft, review, and editing.

### Conflict of Interest Statement

D Colomer reports grants and personal fees from H3 Biomedicine, Novartis, Incyte, AstraZeneca, and AbbVie.

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
