## [Reviewer comments · Life Science Alliance]

Life Science Alliance

SF3B1 Mutation-mediated Sensitization to H3B-8800 Splicing Inhibitor in Chronic Lymphocytic Leukemia

Dolors Colomer, Irene Oreja, Andre Gohr, Heribert Playa-Albinyana, Ariadna Giro, Fabian Arenas, Morihiko Higashi, Rupal tripathi, Monica Lopez-Guerra, Manuel Irimia, Marta Aymerich, Juan Valcarcel, and Sophie Bonnal

DOI: <https://doi.org/10.26508/lsa.202301955>

Corresponding author(s): Dolors Colomer, Consorci Institut D'Investigacions Biomediques August Pi I Sunyer; Sophie Bonnal, Centre for Genomic Regulation (CRG), The Barcelona Institute of Science and Technology and Universitat Pompeu Fabra (UPF), Barcelona, Spain; and Juan Valcarcel, Centre for Genomic Regulation

Review Timeline:

Submission Date:	2023-01-27
Editorial Decision:	2023-02-27
Revision Received:	2023-06-27
Editorial Decision:	2023-07-24
Revision Received:	2023-07-30
Accepted:	2023-07-31

Scientific Editor: Novella Guidi

Transaction Report:

February 27, 2023

Re: Life Science Alliance manuscript #LSA-2023-01955-T

Dr. Dolores Colomer
Institut d'Investigacions Biomèdiques August Pi I Sunyer (IDIBAPS)
Hematopathology Unit. Hospital Clinic
VILLARROEL 170
Barcelona 8036
Spain

Dear Dr. Colomer,

Thank you for submitting your manuscript entitled "SF3B1 Mutation-mediated Sensitization to H3B-8800 Splicing Inhibitor in Chronic Lymphocytic Leukemia" to Life Science Alliance. The manuscript was assessed by expert reviewers, whose comments are appended to this letter. We invite you to submit a revised manuscript addressing the Reviewer comments.

Thank you for this interesting contribution to Life Science Alliance. We are looking forward to receiving your revised manuscript.

Sincerely,

B. MANUSCRIPT ORGANIZATION AND FORMATTING:

Reviewer #1 (Comments to the Authors (Required)):

This is a very comprehensive and well done study of the impact of mutations in SF3B1 on RNA splicing in chronic lymphocytic leukemia (CLL) as well as the impact of the SF3B inhibitor H3B-8800 in preclinical CLL models. The only limitation to the work is that many of the conceptual points have been presented in prior publications. At the same time, the data here utilize a clinical compound (H3B-8800) in the context of CLL specifically and demonstrate preferential activity in SF3B1 mutant CLL which is absent from the literature to date and could be important from a translational perspective. The authors also extend these data to show synergistic combination with venetoclax.

The only aspect of this work which could be presented in more detail is if distinct SF3B1 mutations encountered in CLL could have their own unique as well as shared RNA splicing changes/gene expression effects.

Reviewer #2 (Comments to the Authors (Required)):

In this study, López-Oreja et al. extensively characterized the effects of SF3B1 mutations on RNA missplicing in chronic lymphocytic leukemia (CLL) and investigated the cytotoxic effects of the splicing modulator H3B-8800 in CLL cells with wild type or mutant SF3B1. The authors first used CRISPR/Cas9 genome-editing to develop isogenic MEC1 CLL cell lines carrying the most common mutation (K700E) in one of the two alleles of SF3B1, thereby generating the first CLL cell model for study on the mechanism and consequences of SF3B1 mutations in CLL. Analysis of the RNA-seq data from tumor B cell samples from CLL patients and the genome-edited MEC1 CLL cell lines confirmed that these cell lines recapitulated the RNA missplicing induced by SF3B1 mutations in patient samples, and revealed sequence features associated with the selection of alternative and cryptic 3' splice sites (3'ss) specific to SF3B1 mutations. The authors also found that cryptic 3'ss can be located both upstream and downstream of the canonical 3'ss and that the distance between them can be longer than previously thought. They then experimentally validated a variety of the cryptic 3'ss in the genome-edited MEC1 cell lines to support their computational analyses. Splicing assays using minigenes with mutated branch points showed that activation of downstream cryptic 3'ss in a couple of example genes depends on downstream branch points, adding new insights into the previously proposed mechanism. The authors also identified that one of the target cryptic 3'ss of mutant SF3B1 can be used as a splicing biomarker for SF3B1 mutations in CLL. From the differentially spliced genes, the authors found a few genes involved in the activation of MAPKK and the NF- κ B pathway including one encoding the kinase MAP3K7, of which the missplicing led to reduced protein expression. As a result, the authors detected an increase of the active form of the NF- κ B subunit p65 in SF3B1-K700E MEC1 CLL cell lines. To elucidate the pathogenicity of SF3B1 mutations in CLL, the authors examined the cytotoxic effects of splicing modulator H3B-8800 for the first time in CLL cells. They found that H3B-8800 reduced viability of both wild type and mutant SF3B1 CLL cells in culture with a preferential cytotoxic effect on SF3B1-mutant cells. Xenograft implantation of genome-edited MEC1 CLL cells in immunodeficient mice showed that H3B-8800 inhibited the infiltration of the CLL cells into the bone marrow and spleen. RNA-seq analysis revealed that H3B-8800 induced retention of short introns with high GC contents and skipping of short exons harboring weak splice sites. A number of the disease-relevant splicing changes induced by H3B-8800 were identified in genes involved in apoptosis. This result led the authors to examine the combined effect of H3B-8800 with venetoclax (an inhibitor of the anti-apoptotic protein BCL2) and they found that H3B-8800 showed a synergistic effect with venetoclax on cytotoxicity against MEC1 CLL cells.

Overall, the authors performed a comprehensive analysis of the RNA missplicing regulated by SF3B1 mutations in CLL and revealed detailed sequence features associated with the splicing misregulation, providing new insights into the mechanism. This study also examined for the first time the cytotoxic effects of splicing modulator H3B-8800 in CLL cells with wild type or mutant SF3B1. The manuscript is very well written. The experiments are well done and the results clearly presented. The data in this study are very significant. I only have some minor comments:

1. The authors showed that they identified 863 alternative 3'ss and 268 cryptic 3'ss regulated by SF3B1 mutations. However, they did not provide the actual data containing the information of all these differentially spliced targets, such as the genomic coordinates and PSI values of the misspliced 3'ss. It is important that the authors provide these data in the supplement.
2. Fig. 4F, the authors drew their conclusion based on splicing assays with minigenes harboring mutations of the predicted

branch points. They did not identify the actual branch points by experimental mapping. The BP1 they predicted is located 32-34 nt upstream of the canonical 3'ss, which is typically responsible for the splicing of the upstream cryptic 3'ss according to previous studies. And the results of this study indeed showed that BP1 is responsible for the splicing of the upstream cryptic 3'ss in both wild type and mutant SF3B1 cells. However, the branch point responsible for the splicing of the downstream canonical 3'ss is still unknown. Since this is a very important mechanistic question, the authors need to experimentally map this branch point in wild type MEC1 CLL cells and redo the splicing assay to draw a solid conclusion.

3. Fig. 6A, the RT-PCR results of MAP3K7 and NFKB1 are already shown as subpanels in Fig. 5A. These redundant data should be removed from Fig. 6A.

4. Figs. 6E and 6F, it would help if the authors could show the actual p values in these two panels. On average, the levels of the active form of p65 were increased in mutant SF3B1 cells vs. wild type cells. However, these increases did not reach statistical significance. Since the significance level is arbitrarily set, the actual p values can still be very informative, especially if they are small.

5. According to this study, the mutant-SF3B1 MEC1 CLL cell lines grow more slowly than wild type cells both in culture and in xenograft mouse model. These results seem to contradict with the worse prognosis of CLL patients associated with SF3B1 mutations. It would be inspiring that the authors could speculate a possible explanation in the Discussion section.

Reviewer #3 (Comments to the Authors (Required)):

In this study the authors study the impact genome wide of the mutations of splicing factor SF3B1 which has been involved in several cancers, including chronic lymphocytic leukemia (CLL), and test the impact of specific inhibitors on splicing and cytotoxicity. They use RNA sequencing data from CLL tumor samples, as well as isogenic SF3B1 wild-type and K700E-mutated CLL cell lines that they construct in the study and describe RNA sequences or contexts associated with the changes in splicing patterns, as well as identify specific cryptic splice sites activated in the SF3B1 mutant that can be used as surrogate markers for SF3B1 mutations in CLL. Some of the alternative splicing events induced by the K700E mutation are mechanistically relevant to the activation of the NF- κ B signaling pathway. They study the effects of the H3B-8800 inhibitors on growth and infiltration of WT and K700E mutant cell lines and the effect of treatment on splicing, and they show that the H3B-8800 has synergistic effect with venetoclax .

I am not a specialist of SF3B1 mutations in diseases and the paper is extremely lengthy and dense, with an avalanche of data presented, so it was very difficult for me to read and evaluate it critically. I felt overwhelmed with the amount of data presented, which made a critical reading very difficult. I think the main issue of the paper is that they authors fail to really highlight the novel points revealed by their studies as opposed to just repeating or confirming some of the prior work. The other issue is that the paper felt like a collection of experiments put together in a manuscript without a real feel of unity or direction in the paper, each paragraph representing a different subproject. It is probably a good resource for specialists in the field, but it was very hard to digest for the non-specialist.

I would suggest a global effort of rewriting to make the paper easier to read and also to highlight the really novel points as opposed to confirmatory data.

Specific minor points:

MUT1 cell line appear to exhibit much lower viability than the two other cell lines. Please comment or explain.

On P5 the sentence "Events involving a pair of 3'ss were further classified as alternative or cryptic Alt3'ss, as alternatives to the canonical -the most used in SF3B1WT samples- 3'ss" is very difficult to understand. Please rephrase.

The authors seem to use the word "canonical" splice site (or AG) to indicate the primary splice site used in WT cells. The word "canonical" is generally used to indicates a normal sequence, such that a non-canonical splice site is typically a splice site whose sequence differs from the consensus. So it is very awkward to read "Non-canonical AGs" when the authors describe the main 3'SS. I would suggest to globally replace "canonical" by "primary".

P5 "Alternative and, especially, cryptic 3'ss are generally weaker 3'ss than their canonical counterparts, regardless of their relative location, and display shorter BP to AG distances (Fig.4C and Fig.S2A)"

I am guessing that the actual branchpoints were never mapped for these alternative or cryptic 3'SS, so the term "predicted BP" should be used here.

P9 The two sentences:

"Interestingly, compared to alternative non-changing and cryptic exons, skipped exons display stronger 5'ss and 3'ss and the

surrounding introns and exons are shorter"

AND

"compared to constitutive exons, skipped exons display weaker 5'ss and 3'ss and the surrounding introns are longer"

appear somewhat contradictory and do not explain well the different populations compared. . Please rephrase/explain better.

Reply to reviewers' comments

We would like to thank the reviewers for their thorough review of our manuscript and for the very helpful comments and suggestions, which we feel have considerably improved the presentation of the data and reinforced our conclusions.

Please find below our point-by-point response to their comments, by which we have done our best to carefully address the issues raised by providing new data and analyses as well as by improving the presentation of our results and conclusions. Reviewer's comments are in black and our responses in blue. **All changes and additions to the text and figures are highlighted in red.**

Reviewer #1 (Comments to the Authors (Required)):

This is a very comprehensive and well done study of the impact of mutations in SF3B1 on RNA splicing in chronic lymphocytic leukemia (CLL) as well as the impact of the SF3B inhibitor H3B-8800 in preclinical CLL models. The only limitation to the work is that many of the conceptual points have been presented in prior publications. At the same time, the data here utilize a clinical compound (H3B-8800) in the context of CLL specifically and demonstrate preferential activity in SF3B1 mutant CLL which is absent from the literature to date and could be important from a translational perspective. The authors also extend these data to show synergistic combination with venetoclax.

The only aspect of this work which could be presented in more detail is if distinct SF3B1 mutations encountered in CLL could have their own unique as well as shared RNA splicing changes/gene expression effects.

Our response: We very much appreciate the positive opinion of this reviewer. To address his/her only point, we carried out several types of analyses using transcriptome data from primary samples harboring different mutations in *SF3B1* (8 K700E, 3 H662D, 2 R625C, 1 L743F and 1 T663I). These included PCA as well as unsupervised hierarchical clustering analyses, based upon either differential alternative splicing events or based upon gene expression profiles. PCA analyses did not show clear grouping of different mutations, not even for the more represented K700E mutant, either in terms of alternative splicing (Fig S1A) or gene expression (Fig S1C). Similar conclusions were obtained using clustering analyses (Figs S1B and D). We conclude that the different *SF3B1* mutations are not associated with unique global transcriptomic changes. We next we asked whether specific subsets of gene expression / alternative splicing events could be associated with specific mutations. Because of the limitation of the number of replicates for mutations other than K700E, we compared a group of 8 K700E samples with a group of 3 samples harboring the *SF3B1* H662D mutation. These analyses revealed 136 alternative splicing events (53 IR, 42 Alt3'ss, 33 SE and 8 Alt5'ss), as well as 162 differentially expressed genes (Fig S1E), that can distinguish K700E mutant samples from H662D mutant samples. These results suggest that a specific mutation can lead to particular expression/splicing signatures and -potentially- to unique biological/pathological effects, although larger sample sizes would be necessary to generalize these conclusions. We have included these results in Fig S1 and added this sentence in the text: "PCA analysis and hierarchical clustering in *SF3B1* mutated samples did not reveal global transcriptomic changes associated with specific mutations, although the comparison between 8 *SF3B1* K700E samples and 3 *SF3B1* H662D mutations did identify 136 differential alternative splicing events (53 IR, 42 Alt3'ss, 33 SE and 8 Alt5'ss), as well as 162 differentially expressed genes, suggesting specific molecular effects of

individual mutations (Fig S1).”

Reviewer #2 (Comments to the Authors (Required)):

In this study, López-Oreja et al. extensively characterized the effects of SF3B1 mutations on RNA missplicing in chronic lymphocytic leukemia (CLL) and investigated the cytotoxic effects of the splicing modulator H3B-8800 in CLL cells with wild type or mutant SF3B1. The authors first used CRISPR/Cas9 genome-editing to develop isogenic MEC1 CLL cell lines carrying the most common mutation (K700E) in one of the two alleles of SF3B1, thereby generating the first CLL cell model for study on the mechanism and consequences of SF3B1 mutations in CLL. Analysis of the RNA-seq data from tumor B cell samples from CLL patients and the genome-edited MEC1 CLL cell lines confirmed that these cell lines recapitulated the RNA missplicing induced by SF3B1 mutations in patient samples, and revealed sequence features associated with the selection of alternative and cryptic 3' splice sites (3'ss) specific to SF3B1 mutations. The authors also found that cryptic 3'ss can be located both upstream and downstream of the canonical 3'ss and that the distance between them can be longer than previously thought. They then experimentally validated a variety of the cryptic 3'ss in the genome-edited MEC1 cell lines to support their computational analyses. Splicing assays using minigenes with mutated branch points showed that activation of downstream cryptic 3'ss in a couple of example genes depends on downstream branch points, adding new insights into the previously proposed mechanism. The authors also identified that one of the target cryptic 3'ss of mutant SF3B1 can be used as a splicing biomarker for SF3B1 mutations in CLL. From the differentially spliced genes, the authors found a few genes involved in the activation of MAPKK and the NF- κ B pathway including one encoding the kinase MAP3K7, of which the missplicing led to reduced protein expression. As a result, the authors detected an increase of the active form of the NF- κ B subunit p65 in SF3B1-K700E MEC1 CLL cell lines. To elucidate the pathogenicity of SF3B1 mutations in CLL, the authors examined the cytotoxic effects of splicing modulator H3B-8800 for the first time in CLL cells. They found that H3B-8800 reduced viability of both wild type and mutant SF3B1 CLL cells in culture with a preferential cytotoxic effect on SF3B1-mutant cells. Xenograft implantation of genome-edited MEC1 CLL cells in immunodeficient mice showed that H3B-8800 inhibited the infiltration of the CLL cells into the bone marrow and spleen. RNA-seq analysis revealed that H3B-8800 induced retention of short introns with high GC contents and skipping of short exons harboring weak splice sites. A number of the disease-relevant splicing changes induced by H3B-8800 were identified in genes involved in apoptosis. This result led the authors to examine the combined effect of H3B-8800 with venetoclax (an inhibitor of the anti-apoptotic protein BCL2) and they found that H3B-8800 showed a synergistic effect with venetoclax on cytotoxicity against MEC1 CLL cells.

Overall, the authors performed a comprehensive analysis of the RNA missplicing regulated by SF3B1 mutations in CLL and revealed detailed sequence features associated with the splicing misregulation, providing new insights into the mechanism. This study also examined for the first time the cytotoxic effects of splicing modulator H3B-8800 in CLL cells with wild type or mutant SF3B1. The manuscript is very well written. The experiments are well done and the results clearly presented. The data in this study are very significant. I only have some minor comments:

Our response: We sincerely thank the reviewer for the detailed and very positive feedback on our manuscript. Responses to specific queries are provided below.

1. The authors showed that they identified 863 alternative 3'ss and 268 cryptic 3'ss regulated by SF3B1 mutations. However, they did not provide the actual data containing the information of all these differentially spliced targets, such as the genomic coordinates and PSI values of the misspliced 3'ss. It is important that the authors provide these data in the supplement.

Our response: We are now including the raw data after processing of vast-tools (Figure 3) and Juncexplorer (Figure 4) pipelines, thus providing the identity of all the differentially spliced targets. We also provide the outputs after filtering these tables for patients and cell lines samples. These are shown in

Tables S4-S9. In particular, the data requested by the reviewer can be found in Table S8 and S9.

2. Fig. 4F, the authors drew their conclusion based on splicing assays with minigenes harboring mutations of the predicted branch points. They did not identify the actual branch points by experimental mapping. The BP1 they predicted is located 32-34 nt upstream of the canonical 3'ss, which is typically responsible for the splicing of the upstream cryptic 3'ss according to previous studies. And the results of this study indeed showed that BP1 is responsible for the splicing of the upstream cryptic 3'ss in both wild type and mutant SF3B1 cells. However, the branch point responsible for the splicing of the downstream canonical 3'ss is still unknown. Since this is a very important mechanistic question, the authors need to experimentally map this branch point in wild type MEC1 CLL cells and redo the splicing assay to draw a solid conclusion.

Our response: When working on the study of the various branchpoints, we identified a previously unnoticed nucleotide insertion at position +2 in the sequence of MAP3K7 BP1, MAP3K7BP2, TARBP1BP1 and TARBP2 minigenes used in the previous version of the manuscript, for which we apologize. To set the record straight, we carried out in parallel assays with MAP3K7 and TARBP1 minigenes harboring or not these unexpected mutations. As shown in Figure 1 for the reviewer, while the mutation did not alter the pattern of splicing of MAP3K7 minigenes, it did for TARBP1 minigenes. In the revised version of the manuscript, we have obviously included only the results obtained with the correct constructs. While the conclusions have not changed regarding the experiments involving MAP3K7 minigenes, we have introduced the following change in the text regarding TARBP1: “In the case of the increased relative use of an alternative downstream NAGNAG site in TARBP1, the 3'-most BP appears to be relevant, because its mutation (BP1 mutant) increased the use of canonical 3'ss and prevented activation of the downstream 3' ss. In contrast, the 5'-most BP (BP2) does not seem to have an impact either on the use of the canonical 3'ss or on the use of the downstream 3' ss in the NAGNAG sequence (Fig.4G).”

Figure 1 for reviewers.

(A-B) Sequences of wild type and mutant Alt3'ss regions corresponding to MAP3K7 and TARBP1 included in the minigene constructs with predicted BPs in grey, Alt3'ss in red and Can3'ss in blue. The mistakenly duplicated nucleotide is in green. Single point mutations in BP are indicated in lower case characters. **A. Lower-left panel:** Results of RT-PCR analysis of RNAs from *SF3B1*^{WT} or *SF3B1*^{K700E} MEC1 cell lines transfected with the indicated MAP3K7 minigenes, RNAs purified and analyzed by RT-PCR and polyacrylamide gel electrophoresis. **Lower-right panel:** quantification of the ratio between Alt3'ss and canonical 3'ss usage of the amplification products shown in the left panel. The product corresponding to intron retention migrates at 287 nucleotides, while products corresponding to the usage of alternative 3'ss are annotated in blue (canonical 3'ss) and red (alternative 3'ss) with their respective sizes in the left panels. **B. Lower-left panel:** Fragment analysis of RT-PCR products from RNAs of *SF3B1*^{WT} or *SF3B1*^{K700E} MEC1 cell lines transfected with the indicated TARBP1 minigenes. **Lower-right panel:** ratio between the Alt3'ss and canonical 3'ss PSI quantification (area under the curve of the peaks (e.g. Alt3'ss PSI: aberrant peak/(aberrant peak + canonical peak)) shown in the left panel. BP1 inc: BP1 incorrect; BP2 inc: BP2 incorrect; BP1 cor: BP1 correct; BP2 cor: BP2 correct. Green bars: *SF3B1*^{WT} cells. Orange bars: *SF3B1*^{K700E} cells. TARBP1 minigene replicates: WT = 9, BP1 inc = 6, BP2 inc = 6, BP1 cor = 6, BP2 cor = 6. MAP3K7 minigene replicates: WT = 9, BP1 inc = 6, BP2 inc = 6, BP1 cor = 6, BP2 cor = 6. Kruskal-Wallis test (*: p<0.05; **: p <0.01).

To identify the MAP3K7 canonical 3'ss branchpoint, as requested by the reviewer, we designed multiple new minigenes harboring mutations in one or more of the predicted branch sites in the region (Table S11 and Fig S5). These included the BP that was predicted by lariat profiling to be used with the canonical 3'ss (Zeng et al., 2022), a BP identified by (Li et al., 2021) (BP4), as well as a more systematic mutagenesis of other three sequences matching -to reasonable extent- BP (UNA) sequences. This generated BP1, BP2, BP3, BP4 and BP5 mutants, as well as various combinations of these mutations (BP1.5, BP1.2.3.4 and BP1.2.3.4.5). Strikingly, none of these mutations or combinations compromised splicing at the canonical AG (FigS5 A-C). Of note, a recent report (Li et al., 2021) using a different reporter of the same 3' ss, did not manage to identify specific BP sequences being used either, but rather highlighted that distances between AGs were critical for splice site choice.

We next considered the possibility that a non-canonical sequence could be used as the branch site. To address this, we used RT-PCR across lariat junctions, followed by sequencing of the amplification products (Vogel et al., 1997; Coombes and Boeke, 2005) to map the lariat location in transcripts generated from the WT, BP1 and BP2 minigenes, either transfected in *SF3B1* WT or in K700E MEC1 cell lines. However, despite significant efforts, and as previously reported for this technology (Vogel et al., 1997; Coombes and Boeke, 2005), the combination of inefficient reverse transcription across lariats and PCR amplification artefacts made the results inconclusive. Therefore, despite our best efforts using extensive mutagenesis and lariat mapping, we have not managed to identify the BP used by the canonical 3' ss.

3. Fig. 6A, the RT-PCR results of MAP3K7 and NFKB1 are already shown as subpanels in Fig. 5A. These redundant data should be removed from Fig. 6A.

Our response: We have followed the reviewer's suggestion and removed the RT-PCR results of MAP3K7 and NFKB1 from Fig 5 (before Fig6), which are now shown as Fig S3A.

4. Figs. 6E and 6F, it would help if the authors could show the actual p values in these two panels. On average, the levels of the active form of p65 were increased in mutant SF3B1 cells vs. wild type cells. However, these increases did not reach statistical significance. Since the significance level is arbitrarily set, the actual p values can still be very informative, especially if they are small.

Our response: We have added the p values to panels 5D and 5E (before 6E and 6F).

5. According to this study, the mutant-SF3B1 MEC1 CLL cell lines grow more slowly than wild type cells both in culture and in xenograft mouse model. These results seem to contradict with the worse prognosis of CLL patients associated with SF3B1 mutations. It would be inspiring that the authors could speculate a possible explanation in the Discussion section.

Our response: Following the reviewer's advice, we now include the following comment in the 3rd paragraph of Discussion: "As *SF3B1* mutations are associated with worse prognosis for CLL patients, the lower viability of *SF3B1* K700E MEC1 cell lines compared to WT isogenic cells appears counterintuitive. It should be kept in mind, however, that *SF3B1* mutations have been associated with cellular senescence in murine models and that CLL-like disease only develops in the context of associated *ATM* mutations (Yin et al., 2019). Therefore it seems likely that such additional mutational events are required in combination with *SF3B1* mutations to produce an aggressive cancer phenotype".

Reviewer #3 (Comments to the Authors (Required)):

In this study the authors study the impact genome wide of the mutations of splicing factor SF3B1 which has been involved in several cancers, including chronic lymphocytic leukemia (CLL), and test the impact of specific inhibitors on splicing and cytotoxicity. They use RNA sequencing data from CLL tumor samples, as well as isogenic SF3B1 wild-type and K700E-mutated CLL cell lines that they construct in the study and describe RNA sequences or contexts associated with the changes in splicing patterns, as well as identify specific cryptic splice sites activated in the SF3B1 mutant that can be used as surrogate markers for SF3B1 mutations in CLL. Some of the alternative splicing events induced by the K700E mutation are mechanistically relevant to the activation of the NF- κ B signaling pathway. They study the effects of the H3B-8800 inhibitors on growth and infiltration of WT and K700Emutant cell lines and the effect of treatment on splicing, and they show that the H3B-8800 has synergistic effect with venetoclax.

I am not a specialist of SF3B1 mutations in diseases and the paper is extremely lengthy and dense, with an avalanche of data presented, so it was very difficult for me to read and evaluate it critically. I felt overwhelmed with the amount of data presented, which made a critical reading very difficult. I think the main issue of the paper is that they authors fail to really highlight the novel points revealed by their studies as opposed to just repeating or confirming some of the prior work. The other issue is that the paper felt like a collection of experiments put together in a manuscript without a real feel of unity or direction in the paper, each paragraph representing a different subproject. It is probably a good resource for specialists in the field, but it was very hard to digest for the non-specialist. I would suggest a global effort of rewriting to make the paper easier to read and also to highlight the really novel points as opposed to confirmatory data.

Our response: we appreciate the reviewer's notice of the large volume of data presented and his/her candid opinion of the presentation issues. Following the advice, we have done our best to make the manuscript easier to read and highlight the novelty of our findings relative to previous reports. Specifically, we emphasize 1) the first generation of isogenic genome-edited CLL cell lines differing only by the presence or absence of a single amino acid substitution in SF3B1, 2) the identification of a wider variety of 3' splice site activation events upon SF3B1 mutation compared to previous reports, 3) evidence that H3B-8800 displays therapeutic effects in mouse models of CLL, particularly when used on CLL cells harboring SF3B1 mutations, and 4) evidence of synergism between a SF3B1-targeting drug and venetoclax.

Specific minor points: MUT1 cell line appear to exhibit much lower viability than the two other cell lines. Please comment or explain.

Our response: Following the suggestion of Reviewers 2 and 3, we now include the following comment in

the 3rd paragraph of Discussion: “As *SF3B1* mutations are associated with worse prognosis for CLL patients, the lower viability of *SF3B1* K700E MEC1 cell lines compared to WT isogenic cells appears counterintuitive. It should be kept in mind, however, that *SF3B1* mutations have been associated with cellular senescence in murine models and that CLL-like disease only develops in the context of associated *ATM* mutations (Yin et al., 2019). Therefore it seems likely that such additional mutational events are required in combination with *SF3B1* mutations to produce an aggressive cancer phenotype”.

On P5 the sentence "Events involving a pair of 3'ss were further classified as alternative or cryptic Alt3'ss, as alternatives to the canonical -the most used in SF3B1WT samples- 3'ss" is very difficult to understand. Please rephrase.

Our response: Following the reviewer's advice, we rephrased the sentence as follows: “3'ss choices were classified as “canonical” when they corresponded to the most used 3'ss in *SF3B1*^{WT} samples, while other 3'ss were classified as alternative or cryptic Alt3'ss, depending on whether their use (PSI) in *SF3B1*^{WT} samples was > 5 (alternative Alt3'ss) or ≤ 5 (cryptic Alt3'ss). A change in 3'ss usage $|\Delta\text{PSI}| \geq 15$ upon *SF3B1* mutation was required for the definition of both alternative and cryptic Alt3'ss.”

The authors seem to use the word "canonical" splice site (or AG) to indicate the primary splice site used in WT cells. The word "canonical" is generally used to indicate a normal sequence, such that a non-canonical splice site is typically a splice site whose sequence differs from the consensus. So it is very awkward to read "Non-canonical AGs" when the authors describe the main 3'SS. I would suggest to globally replace "canonical" by "primary".

Our response: we understand the potential confusion of terms raised by the reviewer. However, given that the term “canonical” has been used in this context by previous publications to designate the 3' splice site preferentially used when SF3B1 is wild type (DeBoever et al., 2015; Darman et al., 2015; Wang et al., 2016; Agrawal et al., 2017) we are reluctant to depart from this nomenclature, as it could make it more difficult to compare between studies. We have however done our best to clarify in the text what we mean by these terms in each instance of 3' splice site selection.

P5 "Alternative and, especially, cryptic 3'ss are generally weaker 3'ss than their canonical counterparts, regardless of their relative location, and display shorter BP to AG distances (Fig.4C and Fig.S2A)" I am guessing that the actual branchpoints were never mapped for these alternative or cryptic 3'SS, so the term "predicted BP" should be used here.

Our response: Following the reviewer recommendation, we now use “predicted BP” in this sentence.

P9 The two sentences: "Interestingly, compared to alternative non-changing and cryptic exons, skipped exons display stronger 5'ss and 3'ss and the surrounding introns and exons are shorter" AND "compared to constitutive exons, skipped exons display weaker 5'ss and 3'ss and the surrounding introns are longer" appear somewhat contradictory and do not explain well the different populations compared. Please rephrase/explain better.

Our response: We had defined the control groups in Fig 8C, but following the reviewer's suggestion, we have now added the definition of the control groups in Methods: “The group of regulated exons ($|\Delta\text{PSI}| \geq 15$ and $\text{PSI range} < 5$ upon H3B-8800 treatment) was compared to different control groups (constitutive exons: $\text{PSI} > 95$ in both groups ($n=4922$); alternative non-changing exons in at least one group: $10 < \text{PSI} < 90$ and $|\Delta\text{PSI}| \leq 5$ ($n=3288$); and cryptic exons: $\text{PSI} < 5$ in both groups ($n=4976$)). In the results section, in the sentence where the control groups are mentioned, we added the reference to the Methods and to Fig 8C: “We first compared the group of regulated exons with different control groups (Fig 8C and Methods).”

July 24, 2023

RE: Life Science Alliance Manuscript #LSA-2023-01955-TR

Dr. Dolors Colomer
Consorci Institut D'Investigacions Biomediques August Pi I Sunyer
Hematopathology Unit. Hospital Clinic
VILLARROEL 170
Barcelona 8036
Spain

Dear Dr. Colomer,

Thank you for submitting your revised manuscript entitled "SF3B1 Mutation-mediated Sensitization to H3B-8800 Splicing Inhibitor in Chronic Lymphocytic Leukemia". We would be happy to publish your paper in Life Science Alliance pending final revisions necessary to meet our formatting guidelines.

- please add ORCID ID for secondary and third corresponding authors--they should have received instructions on how to do so
- please add the Twitter handle of your host institute/organization as well as your own or/and one of the authors in our system
- please be sure that all authors are mentioned in the Authors' Contribution section
- please use the [10 author names et al.] format in your references (i.e., limit the author names to the first 10)
- please make sure the manuscript sections are aligned with LSA's formatting guidelines: please separate the Figure legends and Supplemental Figure legends into separate sections
- please add your table legends to the main manuscript text after the legends for supplementary figures
- please add callouts for Figures 7I, S1A-E, S3B, S6A-B to your main manuscript text

A. FINAL FILES:

B. MANUSCRIPT ORGANIZATION AND FORMATTING:

Sincerely,

Reviewer #2 (Comments to the Authors (Required)):

In this revised version of the manuscript, the authors have satisfactorily addressed all of my comments. The manuscript is suitable for publication in its current form.

Reviewer #3 (Comments to the Authors (Required)):

The revised version of the manuscript has addressed the issues raised by the three reviewers. In particular the authors added a transcriptomic comparative analysis of the K700E vs H662D mutations. While they did not identify any significant major transcriptome changes, they identified 136 differential alternative splicing events and 162 differentially expressed genes. These results strengthened the previous data. The authors also identified unbeknownst mutations in the reporters they used in the previous version of the manuscript and had to regenerate the data with the correct reporters. The other issues raised by each reviewer have been addressed in the point-by-point response. I believe that the revised version is acceptable for publication.

July 31, 2023

RE: Life Science Alliance Manuscript #LSA-2023-01955-TRR

Dr. Dolors Colomer
Consorci Institut D'Investigacions Biomediques August Pi I Sunyer
Hematopathology Unit. Hospital Clinic
VILLARROEL 170
Barcelona 8036
Spain

Dear Dr. Colomer,

Thank you for submitting your Research Article entitled "SF3B1 Mutation-mediated Sensitization to H3B-8800 Splicing Inhibitor in Chronic Lymphocytic Leukemia". It is a pleasure to let you know that your manuscript is now accepted for publication in Life Science Alliance. Congratulations on this interesting work.

DISTRIBUTION OF MATERIALS:

Again, congratulations on a very nice paper. I hope you found the review process to be constructive and are pleased with how the manuscript was handled editorially. We look forward to future exciting submissions from your lab.

Sincerely,
